# An Easy Synthesis of Monofluorinated Derivatives of Pyrroles from β-Fluoro-β-Nitrostyrenes

**DOI:** 10.3390/molecules26123515

**Published:** 2021-06-09

**Authors:** Alexander S. Aldoshin, Andrey A. Tabolin, Sema L. Ioffe, Valentine G. Nenajdenko

**Affiliations:** 1Department of Chemistry, Lomonosov Moscow State University, Leninskie Gory 1, 119991 Moscow, Russia; aldon2258@mail.ru; 2N. D. Zelinsky Institute of Organic Chemistry, Russian Academy of Sciences, Leninsky Prosp. 47, 119991 Moscow, Russia; tabolin87@mail.ru (A.A.T.); iof@ioc.ac.ru (S.L.I.)

**Keywords:** β-Fluoro-β-nitrostyrene, vinylpyrrole, Michael addition, elimination, kinetics, Hammett equation, electrophilicity index

## Abstract

The catalyst-free conjugate addition of pyrroles to β-Fluoro-β-nitrostyrenes was investigated. The reaction was found to proceed under solvent-free conditions to form 2-(2-Fluoro-2-nitro-1-arylethyl)-1*H*-pyrroles. The effectiveness of this approach was demonstrated through the preparation of a series of the target products in a quantitative yield. The kinetics of a conjugate addition of pyrrole was studied in detail to reveal the substituent effect and activation parameters of the reaction. The subsequent base-induced elimination of nitrous acid afforded a series of novel 2-(2-Fluoro-1-arylvinyl)-1*H*-pyrroles prepared in up to an 85% isolated yield. The two-step sequence herein proposed is an indispensable alternative to a direct reaction with elusive and unstable 1-Fluoroacetylenes.

## 1. Introduction

The pyrrole ring is a constituent subunit of many naturally produced molecules and pharmaceutically active compounds [1]. Pyrrole and its derivatives are widely used as intermediates in the synthesis of pharmaceuticals, agrochemicals, dyes and other valuable organic compounds [2,3,4,5]. Many drugs containing a pyrrole moiety are available in the market or are currently undergoing clinical trials [6]. They exhibit a wide range of biological activities, such as antibacterial, anticoccidial, anti-inflammatory, antipsychotic, anticonvulsant, antifungal, antiviral, anticancer, etc. [7,8]. The conjugate addition of 2-unsubstituted pyrroles to Michael acceptors is a powerful tool used for their further functionalization [9,10,11,12]. Nitroalkenes are highly reactive and attractive Michael acceptors, giving useful compounds for the subsequent transformation of the nitro group [13]. Indeed, the introduction of a 2-Nitroalkyl moiety into the pyrrole nucleus has attracted much research interest. 2-(2-Nitroalkyl)pyrroles were demonstrated to be appealing intermediates for the synthesis of bioactive compounds [14].

On the other hand, the incorporation of a fluorine into a molecule can positively modulate a number of important pharmacokinetic and physicochemical properties of the drug, such as lipophilicity, electrophilicity, conformation, pKa, metabolic and chemical stability, membrane permeability and the binding affinity to a target protein [15,16,17,18]. Indeed, about one-fourth of the currently manufactured agrochemical and pharmaceutical products contain at least one fluorine atom, and their number tends to persistently rise [19]. In this respect, the development of new routes to fluorinated pyrrole derivatives is of great research interest and importance [20,21].

We recently reported an effective and stereoselective method for the preparation of β-Fluoro-β-nitrostyrenes based on the radical nitration of 2-Bromo-2-fluorostyrenes. This process takes place with the simultaneous elimination of bromine and gives the target structures solely as Z isomers in high yields (up to 92%) [22]. These molecules are very attractive building blocks for the construction of numerous monofluorinated molecules [23,24,25,26,27,28,29,30]. This paper is devoted to the development of routes to new classes of monofluorinated derivatives of pyrrole. We report a catalyst-free conjugate addition of pyrroles to β-Fluoro-β-nitrostyrenes, yielding a novel 2-(2-Fluoro-2-nitro-1-arylethyl)-1*H*-pyrroles **3**. In turn, the latter prove to be effective precursors for the synthesis of novel 2-(2-Fluoro-1-arylvinyl)-1*H*-pyrroles **4** by the base-induced elimination of nitrous acid. This two-step sequence opens up a straightforward way to monofluorinated compounds not previously available. The approach reported herein is an indispensable alternative to the direct addition of pyrroles to elusive and unstable 1-Fluoroacetylenes (Scheme 1).

## 2. Results and Discussion

Initially, we studied the conjugated addition of pyrrole **2a** to β-Fluoro-β-nitrostyrenes **1** to prepare a series of novel 2-(2-Fluoro-2-nitro-1-arylethyl)-1*H*-pyrroles **3**. The transformations were carried out in excess of pyrrole, which acted as both reagent and solvent (Scheme 2).

We found that the reaction proceeded very efficiently without any catalyst at room temperature. The reaction took place only at the α-position of pyrrole, affording the corresponding monofluorinated adducts **3** in a quantitative yield. The reaction demonstrated a broad scope in terms of nitrostyrenes **1** (Scheme 2). It can easily be scaled up to a gram scale without a loss of effectiveness. The NMR spectroscopic analysis showed the formation of a diastereoisomeric mixture of products **3** in a ratio of ca. 40:60. The formation of two diastereomers was probably caused by the high acidity of the proton on the carbon bearing the fluorine and nitro groups (the estimated pKa of CH_2_FNO_2_ was around 9.5) [31]. We also studied the reaction with 1,3-*bis*((*Z*)-2-Fluoro-2-nitrovinyl)benzene **1o**. The formation of *bis*-pyrrole adducts **3o** proceeded in a quantitative yield, giving a mixture of four isomers. Due to the partial overlapping of the resonance signals of F, we did not estimate the molar ratio of all the isomers with ^19^F NMR-spectroscopy. All the structures obtained were elucidated by a combination of NMR spectroscopy and HRMS (See Appendix A).

To gain deeper insights into the reaction, we carried out some kinetic studies to estimate the activation parameters and substituent effect. All the kinetic runs were performed under pseudo-first order conditions using a 216-molar excess of 1*H*-pyrrole. Conversions (F) of **1** were measured by ^19^F NMR spectroscopy. The total effective pseudo-first order rate constants *k** were obtained by plotting the experimental values of ln (*C*_0_/*C*) versus time with good correlations (Table 1 and Table 2). The overall second-order rate total constants *k*_total_ were calculated from the effective *k** and initial concentration of pyrrole (Table 1 and Table 2). The individual constants for the minor and major isomers (*k*_min_ and *k*_maj_) were evaluated by the multiplication of *k*_total_ with the molar fractions of the isomers (Table 1 and Table 2). The reactions were found to proceed under the kinetic control, since the isomer ratio remained constant throughout the reactions course (Table 1 and Table 2). First, the activation parameters were estimated for the reaction of nitrostyrene **1g** with pyrrole. The kinetic studies were conducted in the temperature range of 30–90 °C (Figure 1). The calculated rate constants are presented in Table 1.

The activation parameters were estimated for both the reaction pathways (the formation of two diastereomers) by the Eyring equation [32] (1) from the plots of ln(*k*/T) versus 1/T (2–3). The activation enthalpies ($\Delta H^{\neq }$) and entropies (${\Delta S}^{\neq }$) were found to be 51.72 kJ/mol and −183.45 J/mol K for the major isomer and 55.03 kJ/mol and −178.57 J/mol K for the minor isomer.
ln(*k*/T) = ln(k*_b_*/ħ) + ΔS^≠^/*R* − ΔH^≠^/*R*T(1)
ln(*k*_maj_/T) = 1.675 − 6222.5/T (*R* > 0.999)(2)
ln(*k*_min_/T) = 2.271 − 6622.7/T (*R* > 0.999)(3)

Next, we studied the substituent effect. The kinetic curves were obtained for the reactions of pyrrole with differently para-position substituted nitrostyrenes **1** at 50 °C. The rate constants were calculated and summarized in Table 2. It was found that the reaction rate depended on the nature of a substituent on the benzene ring of the nitrostyrenes **1**. The substituent effect was estimated first by plotting the log k against the Hammett constants σ_p_ for the *para*-substituents (Figure 2).

The following linear relationships were obtained:log *k*_total_ = 0.33σ_p_ − 5.00 (*R* = 0.889)(4)
log *k*_min_ = 0.25σ_p_ − 5.45 (*R* = 0.760)(5)
log *k*_maj_ = 0.38σ_p_ − 5.45 (*R* = 0.928)(6)

The positive value of the reaction constant ρ indicates that the reaction is favored by the withdrawal of electron pairs from the reaction site. However, the low value of ρ (0.3 ÷ 0.4) demonstrates a low sensitivity to the substituent influences. As a consequence, this leads to relatively low correlations of the linear relationships obtained (0.76 ÷ 0.93). Indeed, the ratio of the maximum (p-NO_2_-substituted **1l**) and minimum (p-OCH_3_-substituted **1b**) values of k is around 3, whereas, for the Diels–Alder reaction of nitrostyrenes **1** with 2,3-dimethylbutadiene (ρ = 1.00), the same ratio is around 14 [24].

Another parameter frequently used for the estimation of reactivity is a global electrophilicity index (ω). This parameter is in good correlation with σ_p_. The ground-state geometries of nitrostyrenes **1** were optimized using the DFT B3LYP/6-31G* level of theory [24]. The values of ω were evaluated from the HOMO and LUMO energies of nitrostyrenes **1**:
ω = μ^2^/2η = 1/8 (E_HOMO_ + E_LUMO_)^2^/(E_LUMO_ − E_HOMO_) (7)

Next, the linearization of the log k versus ω was made for the formation of both diastereomers (Figure 3). However, the correlations of the linear relationships obtained were worse (8–10).
log *k*_total_ = 0.31ω − 5.90 (*R* = 0.843)(8)
log *k*_min_ = 0.22ω − 6.10 (*R* = 0.700) (9)
log *k*_maj_ = 0.35ω − 6.22 (*R* = 0.887)(10)

Next, we selected a series of nitrostyrenes differing only in a substituent at the double bond to evaluate the substituent effect at β-carbon (Scheme 3). The kinetics studies of reactions of pyrrole with nitrostyrenes bearing H, F, Cl and Me at the terminal carbon were conducted at 50 °C (Figure 4). The data obtained are summarized in Table 3. In this case, a high sensitivity to the substituent influences was observed because of the near position of the substituent to the reaction site. The least reactivity was demonstrated by β-methylated nitrostyrene **1b′** (Table 3, Entry 1). Its conversion was only 17% after 4 h of the reaction. Fluorinated nitrostyrene **1b** showed a 6.5 times faster total reaction rate than that of the methylated one **1b′**.

However, β-unsubstituted nitrostyrene **1b″** having the minimum sterical demand was about 18 times more reactive than **1b′**. To our surprise, β-chlorinated nitrostyrene **1b‴** (Table 3, Entry 4) proved the most reactive one. Almost a full conversion was achieved only after 1 h. It was nearly 69 times more reactive than **1b′**. Such an effect of the substituent at β-carbon can be explained by the combination of electronic and sterical effects.

The novel 2-(2-Fluoro-2-nitro-1-arylethyl)-1*H*-pyrroles **3** obtained are interesting compounds due to their potential synthetic utility for the subsequent transformations. We proposed that they could be appropriate and effective precursors for novel 2-(2-Fluoro-1-arylvinyl)-1*H*-pyrrole **4**. C-Vinylpyrroles have attracted much research interest due to their synthetic utility as building blocks for pyrrole derivatives [33]. However, by now, 2-(2-Fluoro-vinyl)pyrroles have not been obtained yet. Apparently, this is related to the unavailability of suitable precursors. Indeed, 1-Fluoroacetylenes that could be potential precursors for the synthesis of such structures are elusive, unstable compounds. However, the use of β-fluoro-β-nitrostyrenes **1** as synthetic equivalents of 1-fluoroacetylenes is a promising way to overcome this synthetic problem. We applied this strategy based on the formation of a double bond in the final step by the base-induced elimination of nitrous acid from the adducts **3** (Scheme 4).

The reactions were conducted in acetonitrile using DBU as a base at an ambient temperature (Scheme 4). In principle, the competitive elimination of either HNO_2_ or HF can be expected, leading to the formation of either fluorine-substituted **4** or nitro-substituted 2-Vinylpyrrole **5**. However, to our delight, the reaction demonstrated a high selectivity towards the desired 2-(2-Fluoro-1-arylvinyl)-1*H*-pyrroles **4**. The reaction demonstrated a high effectiveness to furnish the desired products **4** in up to 85% yields, whereas the side products **5** did not exceed 15%. Somewhat lower yields (43–50%) were achieved in the case of the adducts **3b** and **3l**, having a strong electron-donating (EDG) methoxy- or a strong electron-withdrawing (EWG) nitro group at the para-position of the benzene ring, correspondingly. The presence of a nitro group at the ortho-position of the benzene substituent of **3n** resulted in the decomposition of the reaction mixture under these reaction conditions. Only trace amounts of adduct **4n** were detected. However, when the nitro-group was posed at the meta-position, a high yield of product **4m** (78%) was obtained. All the other adducts **3** were also successfully transformed into the corresponding 2-vinylpyrroles **4**.

Both the fluorinated product **4** and nitro product **5** were formed as a mixture of *Z*- and *E*-isomers. However, both the fluorinated isomers **4**, if necessary, can be separated by column chromatography on silica gel, whereas, in the case of the side product **5**, both isomers have the same retention time and, therefore, are isolated together. We demonstrated the possibility of separation (*Z*)- and (*E*)-2-(2-Fluoro-1-phenylvinyl)-1*H*-pyrrole for a large number of examples and fully characterized them. As expected, the elimination of the bis-pyrrole adduct **3o** resulted in the formation of the desired difluorinated product **4o** as a mixture of three isomers, with the *Z*,*Z*-, *E*,*Z*- and *E*,*E*- configurations in a 24:48:28 ratio, correspondingly. The side products were monofluorinated nitro-derivatives **5o** formed as a mixture of four isomers. We isolated two mixtures of isomers in the *Z*-NO_2_ and *Z*-F, along with *E*-NO_2_ and *Z*-F configurations and the *Z*-NO_2_ and *E*-F, along with *E*-NO_2_ and *E*-F configurations.

The stereochemistry of the products **4** was unambiguously assigned with NMR spectroscopy. The ^1^H NMR spectra displayed a significant downfield shift (ca. 1 ppm) of pyrrolic H5 in a *Z*-configuration in reference to that in the *E*-configuration (Figure 5). Additionally, the splitting of the resonance signal of fluorine into a double doublet in the *Z*-configuration was observed with ^19^F NMR spectroscopy (Figure 5), whereas, in the *E*-configuration, the fluorine resonance signal was a doublet. Probably, this indicated the formation of an intermolecular hydrogen bond with fluorine. Indeed, a similar pattern was observed for the side nitro products **5**. In this case, the downfield shift of the pyrrolic proton in the *Z*-configuration was much greater ca. 3.7 ppm. The difference in the frequencies of the NH vibrations regarding the *Z*- and *E*-positions of fluorine observed with IR spectroscopy was about 20 cm^−1^ (Figure 6).

As expected, the resulting ratio of the diastereomers **4** remained similar to that of the precursors **3**. However, the elimination of the 2,4-dichlorophenyl-substituted adduct **3b** with *dr* = 48:52 led to the predominant formation of a *Z*-isomer **4** in high diastereomeric excess (90% *de*).

Next, we demonstrated the versatility of our methodology for *N*-substituted pyrroles. Due to the reduced nucleophilicity of *N*-aryl pyrroles and high steric demand of substituents, the conjugate addition required harsher conditions. Moreover, most of *N*-aryl pyrroles used are solid at room temperature (except for pyrrole **2g**) and need heating to be melted. We applied catalyst-free conditions for the reactions of nitrostyrene **1g** with *N*-substituted pyrroles (Scheme 5). The reactions were carried out under solvent-free conditions and microwave-activated at elevated temperatures (150–200 °C) for 16–20 h. This, in turn, intensified the undesirable side reactions and, as a consequence, in some cases, decreased the selectivity. Indeed, some amount of β-substituted adducts was observed along with the target products. The reaction with 1-methyl-1*H*-pyrrole **2b** was completed at 150 °C with a good overall yield of **3p** (76%). However, the formation of 13% of the β-substituted regioisomers was detected by ^1^H and ^19^F-NMR spectroscopy. The reactions of nitrostyrene **1g** with 1-Phenyl-1*H*-pyrrole **2c** and 1-(4-Ethylphenyl)-1*H*-pyrrole **2d** demonstrated poor reactivity. Low yields of adducts **3q** and **3r** (28–35%) were obtained after the full conversion of **1g**. However, the reaction with 1-(p-Tolyl)-1*H*-pyrrole **2e** displayed a better reactivity to give adduct **3s** in a 51% yield. It was also found that the reactivity of *N*-aryl-substituted pyrroles was increased in the presence of an electron-donating methoxy group in the benzene ring. For instance, the reaction with 1-(3-Methoxyphenyl)-1*H*-pyrrole **2g** resulted in a good yield of adduct **3u** (74%), whereas, in the case of the simultaneous presence of EDG (-OMe) and weak EWG (-Cl), the yield of the adduct **3t** (46%) diminished compared to **3u**. On the other hand, we failed to obtain the adducts based on pyrroles having only electron-withdrawing groups onthe aryl-substituent under these conditions.

After, the subsequent elimination of nitrous acid from the adducts **3p**–**3u** was conducted (Scheme 5). Similar to their N-unsubstituted analogs, these compounds also reacted smoothly. The corresponding 2-vinyl pyrroles **4p**–**4s** were obtained in good yields (57–72%). The side nitro product **5** was isolated in the range of 10–18%. It was found that *N*-aryl-substituted 2-vinylpyrroles formed with a predominance of the *Z*-isomer. It can be clearly explained by the steric demand of the aryl substituent with nitrogen compared to the hydrogen or methyl groups. Indeed, in the course of conjugate addition (Scheme 3), the orientation of the small fluorine near a bulky aryl is sterically more favorable.

The pyrrole derivatives are also of great importance for the modification at the free α-position. Therefore, we studied the reaction of adduct **3l** with 4-Chlorobenzaldehyde to prepare the corresponding dipyrromethane **6** (Scheme 6). The reaction proceeded in the presence of trifluoroacetic acid (TFA) as the catalyst and gave **6** as a mixture of the diastereomers in a 48% yield. This transformation can open up the straightforward way to novel valuable classes of fluorinated pyrrole derivatives, such as dipyrromethanes and their boron difluoride complexes (BODIPYs).

## 3. Materials and Methods

All reagents were purchased from commercial sources and used without any further purification. All solvents were dried before use by the standard procedure [34]. Melting points (m.p.) were measured with a Büchi B-545 melting point apparatus (Büchi, Flawil, Switzerland). Microwave activated reactions were conducted in Monowave 200 (Anton Paar, Graz, Austria). NMR (^1^H, ^13^C and ^19^F) spectra were obtained with the Bruker AVANCE 400 (Bruker Corp., Karlsruhe, Germany) and Agilent 400-MR spectrometers (Agilent Technologies, Santa Clara, CA, USA) using deuterated chloroform (CDCl_3_). Chemical shifts for the ^1^H NMR spectroscopic data were referenced to the internal tetramethylsilane (δ = 0.0 ppm) and the residual solvent resonance (δ = 7.26 ppm); chemical shifts for the ^13^C NMR spectroscopic data were referenced to the residual solvent resonance (δ = 77.16 ppm); chemical shifts for the ^19^F NMR spectroscopic data were referenced to PhCF_3_ (δ = −63.72 ppm) or C_6_F_6_ (δ = −164.90 ppm). Data were reported as follows: chemical shift; integration multiplicity (s = singlet, d = doublet, t = triplet, q = quadruplet, qui = quintet, sext = sextet, sept = septet, br = board, m = multiplet, dd = doublet of doublets and ddd = doublet of doublet of doublets) and coupling constants (Hz). Starting β-fluoro-β-nitrostyrenes were prepared according to the described procedures [22,35]. All nitrostyrenes are known compounds [22,23,24,25,26,27,28,29,30].

General procedure for the conjugate addition of 1H-pyrrole to β-fluoro-β-nitrostyrenes.

In a typical experiment, β-fluoro-β-nitrostyrene (0.5 mmol) and pyrrole (0.5 mL) were successively loaded into a vial. The reaction mixture was stirred at room temperature for 25–30 h. After completion of the reaction (TLC monitoring), the excess of pyrrole was evaporated under vacuum. The desired product was isolated by column chromatography on silica gel as a mixture of diastereomers.

**2-(2-Fluoro-2-nitro-1-phenylethyl)-1H-pyrrole** (**3a**). Eluent: Hex/DCM 1:1, DCM; 107 mg (88 %); *dr* = 42:58; colorless oil. ^1^H NMR (400 MHz, CDCl_3_): (major isomer) δ = 7.99 (br s, 1H), 7.46–7.23 (m, 5H), 6.78–6.71 (m, 1H), 6.33–6.28 (m, 1H), 6.28 (dd, ^2^*J*_HF_ = 50.0, 3.5 Hz, 1H), 6.20 (t, *J* = 2.3 Hz, 1H), 4.97 (dd, ^3^*J*_HF_ = 28.5, 3.4 Hz, 1H) ppm; (minor isomer) δ = 8.11 (br s, 1H), 7.46–7.23 (m, 5H), 6.78–6.71 (m, 1H), 6.26–6.23 (m, 1H), 6.20 (t, *J* = 2.3 Hz, 1H), 6.19 (dd, ^2^*J*_HF_ = 50.0, 3.4 Hz, 1H), 4.99 (dd, ^3^*J*_HF_ = 28.0, 4.0 Hz, 1H) ppm. ^13^C NMR (100 MHz, CDCl_3_): (major isomer) δ = 133.2, 129.2, 129.1, 128.4, 125.7, 118.9, 111.5 (d, ^1^*J*_CF_ = 245.4 Hz), 109.0, 107.6, 47.9 (d, ^2^*J*_CF_ = 18.7 Hz) ppm; (minor isomer) δ = 135.1 (d, ^3^*J*_CF_ = 1.6 Hz), 129.4, 128.8, 128.6, 123.8, 119.2, 112.4 (d, ^1^*J*_CF_ = 243.7 Hz), 109.2 (d, ^4^*J*_CF_ = 0.8 Hz), 108.9, 48.0 (d, ^2^*J*_CF_ = 18.6 Hz) ppm; ^19^F NMR (376 MHz, CDCl_3_): (major isomer) δ = −153.26 (dd, *J* = 50.0, 28.5 Hz) ppm; (minor isomer) δ = −151.19 (dd, *J* = 50.0, 28.0 Hz) ppm. HRMS (ESI-TOF) *m*/*z*: [M − H]^−^ calcd. for C_12_H_10_FN_2_O_2_ 233.0732; found 233.0735.

**2-(2-Fluoro-1-(4-methoxyphenyl)-2-nitroethyl)-1H-pyrrole** (**3b**). Eluent: Hex/DCM 1:1, DCM; 67 mg (>99%); *dr* = 40:60; colorless oil. ^1^H NMR (400 MHz, CDCl_3_): (major isomer) δ = 7.99 (br s, 1H), 7.16 (d, *J* = 8.5 Hz, 2H), 6.86 (d, *J* = 8.7 Hz, 2H), 6.77–6.69 (m, 1H), 6.27 (dd, ^2^*J*_HF_ = 50.2, 3.2 Hz, 1H), 6.26 (s, 1H), 6.24–6.14 (m, 1H), 4.92 (dd, ^3^*J*_HF_ = 29.0, 3.5 Hz, 1H), 3.78 (s, 3H) ppm; (minor isomer) δ = 8.11 (br s, 1H), 7.23 (d, *J* = 8.6 Hz, 2H), 6.91 (d, *J* = 8.7 Hz, 2H), 6.77–6.69 (m, 1H), 6.14 (dd, ^2^*J*_HF_ = 50.1, 3.8 Hz, 1H), 6.24–6.13 (m, 2H), 4.91 (dd, ^3^*J*_HF_ = 27.0, 4.3 Hz, 1H), 3.81 (s, 3H) ppm. ^13^C NMR (100 MHz, CDCl_3_): (major isomer) δ = 159.9, 130.4, 126.1, 125.0, 118.7, 114.5, 111.5 (d, ^1^*J*_CF_ = 244.9 Hz), 109.0, 107.3, 55.4, 47.2 (d, ^2^*J*_CF_ = 18.9 Hz) ppm; (minor isomer) δ = 159.7, 129.7, 126.8 (d, ^3^*J*_CF_ = 2.3 Hz), 124.3, 119.0, 114.7, 112.7 (d, ^1^*J*_CF_ = 243.5 Hz), 108.9 (d, ^4^*J*_CF_ = 0.8 Hz), 108.9, 55.4, 47.3 (d, ^2^*J*_CF_ = 18.7 Hz) ppm. ^19^F NMR (376 MHz, CDCl_3_): (major isomer) δ = −153.92 (dd, *J* = 50.2, 29.0 Hz) ppm; (minor isomer) δ = −150.63 (dd, *J* = 50.1, 27.0 Hz) ppm. HRMS (ESI-TOF) *m*/*z*: [M + H]^+^ calcd. for C_13_H_14_FN_2_O_3_ 265.0983; found 265.0986.

**2-(1-(4-(*Tert*-butyl)phenyl)-2-fluoro-2-nitroethyl)-1H-pyrrole** (**3c**). Eluent: Hex/DCM 1:1, DCM; 131 mg (85 %); *dr* = 42:58; yellowish oil. ^1^H NMR (400 MHz, CDCl_3_): (major isomer) δ = 7.96 (br s, 1H), 7.38 (d, *J* = 8.4 Hz, 2H), 7.20 (d, *J* = 8.2 Hz, 2H), 6.75–6.71 (m, 1H), 6.29 (br s, 1H), 6.24 (dd, *J* = 5.8, 3.2 Hz, 1H), 6.27 (dd, ^2^*J*_HF_ = 50.1, 3.6 Hz, 1H), 4.96 (dd, ^3^*J*_HF_ = 28.3, 3.5 Hz, 1H), 1.33 (s, 9H) ppm; (minor isomer) δ = 8.09 (br s, 1H), 7.44 (d, *J* = 8.4 Hz, 2H), 7.27 (d, *J* = 9.0 Hz, 2H), 6.75–6.71 (m, 1H), 6.17–6.21 (m, 2H), 6.18 (dd, ^2^*J*_HF_ = 50.1, 3.7 Hz, 1H), 4.99 (dd, ^3^*J*_HF_ = 28.4, 3.8 Hz, 1H), 1.36 (s, 9H) ppm. ^13^C NMR (100 MHz, CDCl_3_): (major isomer) δ = 151.7, 130.1, 128.9, 126.1, 125.9 (d, ^4^*J*_CF_ = 0.9 Hz), 118.7, 111.6 (d, ^1^*J*_CF_ = 245.1 Hz), 109.0, 107.4, 47.5 (d, ^2^*J*_CF_ = 18.9 Hz), 34.6, 31.3 ppm; (minor isomer) δ = 151.8, 132.1 (d, ^3^*J*_CF_ = 1.3 Hz), 128.1, 126.3, 123.9, 119.0, 112.4 (d, ^1^*J*_CF_ = 243.9 Hz), 109.1, 108.9, 47.5 (d, ^2^*J*_CF_ = 18.4 Hz), 34.7, 31.3 ppm.^19^F NMR (376 MHz, CDCl_3_): (major isomer) δ = −152.75 (dd, *J* = 50.1, 28.3 Hz) ppm; (minor isomer) δ = −151.15 (ddd, *J* = 50.1, 28.4, 5.0 Hz) ppm. HRMS (ESI-TOF) *m*/*z*: [M + H]^+^ calcd. for C_16_H_20_FN_2_O_2_ 291.1509; found 291.1503.

**2-(2-Fluoro-2-nitro-1-(p-tolyl)ethyl)-1H-pyrrole** (**3d**). Eluent: Hex/DCM 1:1; 122 mg (99 %); *dr* = 44:56; yellowish oil. ^1^H NMR (400 MHz, CDCl_3_): (major isomer) δ = 8.09 (br s, 1H), 7.20–7.12 (m, 4H), 6.77–6.70 (m, 1H), 6.31–6.27 (m, 1H), 6.27 (dd, ^2^*J*_HF_ = 50.1, 3.5 Hz, 1H), 6.25 (dd, *J* = 5.7, 2.5 Hz, 1H), 4.94 (dd, ^3^*J*_HF_ = 28.4, 3.4 Hz, 1H), 2.36 (s, 3H) ppm; (minor isomer) δ = 7.97 (br s, 1H), 7.22 (s, 4H), 6.77–6.70 (m, 1H), 6.21–6.17 (m, 2H), 6.17 (dd, ^2^*J*_HF_ = 50.1, 4.2 Hz, 1H), 4.95 (dd, ^3^*J*_HF_ = 27.6, 4.3 Hz, 1H), 2.39 (s, 3H) ppm. ^13^C NMR (100 MHz, CDCl_3_): (major isomer) δ = 138.7, 130.1, 129.8, 129.1, 125.9, 118.7, 111.5 (d, ^1^*J*_CF_ = 245.3 Hz), 109.0, 107.4 (d, ^4^*J*_CF_ = 1.0 Hz), 47.6 (d, ^2^*J*_CF_ = 18.9 Hz), 21.2 ppm; (minor isomer) δ = 138.6 ppm, 131.9 (d, ^3^*J*_CF_ = 2.1 Hz), 130.0, 128.3, 124.1, 119.0, 112.6 (d, ^1^*J*_CF_ = 243.5 Hz), 109.0, 108.8, 47.6 (d, ^2^*J*_CF_ = 18.9 Hz), 21.2 ppm. ^19^F NMR (376 MHz, CDCl_3_): (major isomer) δ = −153.31 (dd, *J* = 50.1, 28.4 Hz) ppm; (minor isomer) δ = −150.74 (dd, *J* = 50.1, 27.6 Hz) ppm. HRMS (ESI-TOF) *m*/*z*: [M + H]^+^ calcd. for C_13_H_14_FN_2_O_2_ 249.1039; found 249.1054.

**2-(2-Fluoro-1-(4-fluorophenyl)-2-nitroethyl)-1H-pyrrole** (**3e**). Eluent: Hex/DCM 1:1, DCM; 118 mg (94%); *dr* = 44:56; yellowish oil. ^1^H NMR (400 MHz, CDCl_3_): (major isomer) δ = 8.01 (br s, 1H), 7.22 (dd, *J* = 8.5, 5.2 Hz, 2H), 7.06 –6.98 (m, 2H), 6.80–6.72 (m, 1H), 6.29 (br s, 1H), 6.28 (dd, ^2^*J*_HF_ = 50.0, 3.2 Hz, 1H), 6.26–6.15 (m, 1H), 4.97 (dd, ^3^*J*_HF_ = 28.8, 3.1 Hz, 1H) ppm; (minor isomer) δ = 8.14 (br s, 1H), 7.30 (dd, *J* = 8.5, 5.2 Hz, 2H), 7.05–7.13 (m, 2H), 6.72–6.80 (m, 1H), 6.15–6.26 (m, 2H), 6.16 (dd, ^2^*J*_HF_ = 50.0, 3.5 Hz, 1H), 4.97 (dd, ^3^*J*_HF_ = 27.6, 3.1 Hz, 1H) ppm. ^13^C NMR (100 MHz, CDCl_3_): (major isomer) δ = 162.9 (d, ^1^*J*_CF_ = 248.2 Hz), 131.0 (d, ^3^*J*_CF_ = 8.2 Hz), 129.0 (d, ^3^*J*_CF_ = 3.1 Hz), 125.5, 119.0, 116.1 (d, ^2^*J*_CF_ = 21.4 Hz), 111.2 (d, ^1^*J*_CF_ = 245.4 Hz), 109.1, 107.7, 47.2 (d, ^2^*J*_CF_ = 18.8 Hz) ppm; (minor isomer) δ = 162.7 (d, ^1^*J*_CF_ = 248.2 Hz), 130.9 (t, ^3^*J*_CF_ = 2.6 Hz), 130.2 (d, ^3^*J*_CF_ = 8.2 Hz), 123.6, 119.3, 116.3 (d, ^2^*J*_CF_ = 21.3 Hz), 47.3 (d, ^2^*J*_CF_ = 18.6 Hz), 109.0, 109.3, 112.3 (d, ^1^*J*_CF_ = 243.7 Hz) ppm. ^19^F NMR (376 MHz, CDCl_3_): (major isomer) δ = −113.92 (tt, *J* = 8.5, 5.2 Hz), −154.01 (dd, *J* = 50.0, 28.8 Hz) ppm; (minor isomer) δ = −114.10 (tt, *J* = 8.5, 5.2 Hz), −151.24 (dd, *J* = 50.0, 27.6 Hz) ppm. HRMS (ESI-TOF) *m*/*z*: [M − H]^−^ calcd. for C_12_H_9_F_2_N_2_O_2_ 251.0638; found 251.0644.

**2-(1-(4-Bromophenyl)-2-fluoro-2-nitroethyl)-1H-pyrrole** (**3f**). Eluent: Hex/DCM 1:1; 133 mg (84 %); *dr* = 44:56; yellowish oil; ^1^H NMR (400 MHz, CDCl_3_): (major isomer) δ = 8.01 (br s, 1H), 7.49–7.44 (m, 2H), 7.11 (d, *J* = 8.3 Hz, 2H), 6.76 (ddd, *J* = 4.1, 2.8, 1.5 Hz, 1H), 6.31–6.27 (m, 1H), 6.27 (dd, ^2^*J*_HF_ = 49.9, 3.3 Hz, 1H), 6.24 (dd, *J* = 6.1, 2.8 Hz, 1H), 4.94 (dd, ^3^*J*_HF_ = 28.6, 3.5 Hz, 1H) ppm; (minor isomer) δ = 8.13 (br s, 1H), 7.55–7.50 (m, 2H), 7.19 (d, *J* = 8.5 Hz, 2H), 6.76 (ddd, *J* = 4.1, 2.8, 1.5 Hz, 1H), 6.16 (dd, ^2^*J*_HF_ = 50.0, 4.0 Hz, 1H), 6.14–6.20 (m, 2H), 4.95 (dd, ^3^*J*_HF_ = 28.2, 5.2 Hz, 1H) ppm. ^13^C NMR (100 MHz, CDCl_3_): δ = 134.1 (d, ^3^*J*_CF_ = 1.8 Hz), 132.5, 132.3, 132.2, 130.1, 125.1, 123.1, 122.8, 119.4, 119.1, 111.9 (d, ^1^*J*_CF_ = 244.0 Hz), 111.0 (d, ^1^*J*_CF_ = 245.7 Hz), 109.5, 109.1, 109.0, 107.9 (d, ^4^*J*_CF_ = 1.2 Hz), 47.4 (d, ^2^*J*_CF_ = 18.7 Hz), 47.3 (d, ^2^*J*_CF_ = 18.7 Hz) ppm.^19^F NMR (376 MHz, CDCl_3_): (major isomer) δ = −153.62 (dd, *J* = 49.9, 28.6 Hz) ppm; (minor isomer) δ = −151.36 (dd, *J* = 50.0, 28.2 Hz) ppm. HRMS (ESI-TOF) *m*/*z*: [M + H]^+^ calcd. for C_12_H_11_^80^BrFN_2_O_2_ 312.9982; found 312.9951.

**2-(1-(4-Chlorophenyl)-2-fluoro-2-nitroethyl)-1H-pyrrole** (**3g**). Eluent: Hex/DCM 1:1, DCM; 131 mg, (97 %); *dr* = 42:58; yellowish oil. ^1^H NMR (400 MHz, CDCl_3_): (major isomer) δ = 8.02 (br s, 1H), 7.32 (d, *J* = 8.4 Hz, 2H), 7.18 (d, *J* = 8.4 Hz, 2H), 6.76 (s, 1H), 6.29 (br s, 1H), 6.27 (dd, *J* = 50.0, 3.3 Hz, 1H), 6.25 (dd, *J* = 5.8, 2.8 Hz, 1H), 4.95 (dd, ^3^*J*_HF_ = 28.6, 3.2 Hz, 1H) ppm; (minor isomer) δ = 8.14 (br s, 1H), 7.37 (d, *J* = 8.4 Hz, 2H), 7.26 (d, *J* = 8.4 Hz, 1H), 6.76 (s, 1H), 6.16 (dd, ^2^*J*_HF_ = 50.0, 4.0 Hz, 2H), 6.21–6.15 (m, 2H), 4.97 (dd, ^3^*J*_HF_ = 27.9, 5.0 Hz, 1H) ppm; ^13^C NMR (100 MHz, CDCl_3_): (major isomer) δ = 134.9, 131.7, 130.5, 129.3, 125.1, 119.1, 111.1 (d, ^1^*J*_CF_ = 245.6 Hz), 109.1, 107.8 (d, ^4^*J*_CF_ = 1.1 Hz), 47.2 (d, ^2^*J*_CF_ = 18.7 Hz) ppm; (minor isomer) δ = 134.6, 133.6 (d, ^3^*J*_CF_ = 1.5 Hz), 129.8, 129.5, 123.2, 119.4, 112.0 (d, ^1^*J*_CF_ = 244.0 Hz), 109.4, 109.0, 47.3 (d, ^2^*J*_CF_ = 18.8 Hz) ppm; ^19^F NMR (376 MHz, CDCl_3_): (major isomer) δ = −153.65 (dd, *J* = 50.0, 28.6 Hz) ppm; (minor isomer) δ = −151.35 (dd, *J* = 50.0, 27.9 Hz) ppm. HRMS (ESI-TOF) *m*/*z*: [M + H]^+^ calcd. for C_12_H_11_^35^ClFN_2_O_2_ 269.0493; found 269.0485.

**2-(1-(2,4-Dichlorophenyl)-2-fluoro-2-nitroethyl)-1H-pyrrole** (**3h**). Eluent: Hex/DCM 1:1; 145 mg, (96 %); *dr* = 48:52; yellowish oil; ^1^H NMR (400 MHz, CDCl_3_): δ = 8.21 (br s, 1H) ppm, 8.11 (br s, 1H), 7.51–7.47 (m, 1H), 7.46–7.41 (m, 2H), 7.29–7.22 (m, 3H), 6.78–6.73 (m, 2H), 6.31 (dd, ^2^*J*_HF_ = 50.0, 3.8 Hz, 1H), 6.29–6.24 (m, 2H), 6.21 (td, *J* = 6.3, 2.8 Hz, 2H), 6.19 (dd, ^2^*J*_HF_ = 49.9, 3.3 Hz, 1H), 5.59 (dd, ^3^*J*_HF_ = 26.5, 4.0 Hz, 1H), 5.57 (dd, ^3^*J*_HF_ = 30.8, 3.2 Hz, 1H) ppm.^13^C NMR (100 MHz, CDCl_3_): δ = 135.3, 135.2, 134.9, 134.0, 131.8, 131.5 (d, *J*_CF_ = 1.0 Hz), 131.1 (d, *J*_CF_ = 4.1 Hz), 130.5, 130.0, 129.9, 128.1, 128.0, 124.3 (d, *J*_CF_ = 1.8 Hz), 121.8, 119.3, 119.2, 110.8 (d, ^1^*J*_CF_ = 246.1 Hz), 110.1 (d, ^1^*J*_CF_ = 244.1 Hz), 110.1 (d, ^4^*J*_CF_ = 2.0 Hz), 109.3, 109.2, 108.3, 44.2 (d, ^2^*J*_CF_ = 18.4 Hz), 42.9 (d, ^2^*J*_CF_ = 18.7 Hz) ppm. ^19^F NMR (376 MHz, CDCl_3_): (major isomer) δ = −155.07 (dd, *J* = 49.8, 30.7 Hz) ppm; (minor isomer) δ = −151.19 (dd, *J* = 49.8, 26.3 Hz) ppm. HRMS (ESI-TOF) *m*/*z*: [M + H]^+^ calcd. for C_12_H_10_^35^Cl_2_FN_2_O_2_ 303.0098; found 303.0095.

**2-(2-Fluoro-2-nitro-1-(4-(trifluoromethyl)phenyl)ethyl)-1H-pyrrole** (**3i**). Eluent: Hex/DCM 1:1, DCM; 142 mg (94%); *dr* = 38:62; yellow solid; M.p. = 68–72 ºC. ^1^H NMR (400 MHz, CDCl_3_): (major isomer) δ = 8.04 (br s, 1H), 7.61 (d, *J* = 8.1 Hz, 2H), 7.39 (d, *J* = 8.1 Hz, 2H), 6.81–6.75 (m, 1H), 6.31 (dd, ^2^*J*_HF_ = 49.8, 3.4 Hz, 1H), 6.35–6.30 (m, 1H), 6.21 (t, *J* = 2.4 Hz, 1H), 5.05 (dd, ^3^*J*_HF_ = 28.3, 3.3 Hz, 1H) ppm; (minor isomer) δ = 8.17 (br s, 1H), 7.67 (d, *J* = 8.2 Hz, 2H), 7.46 (d, *J* = 8.1 Hz, 2H), 6.75–6.81 (m, 1H), 6.22–6.30 (m, 1H), 6.22 (dd, ^2^*J*_HF_ = 49.9, 3.7 Hz, 1H), 6.21 (t, *J* = 2.4 Hz, 1H), 5.08 (dd, ^3^*J*_HF_ = 28.6, 3.7 Hz, 1H) ppm. ^13^C NMR (100 MHz, CDCl_3_): (major isomer) δ = 137.4, 131.0 (q, ^2^*J*_CF_ = 32.8 Hz), 129.7, 126.1 (q, ^3^*J*_CF_ = 3.7 Hz), 124.7, 123.9 (q, ^1^*J*_CF_ = 272.2 Hz), 119.4, 111.1 (d, ^1^*J*_CF_ = 245.9 Hz), 109.3, 108.2 (d, ^4^*J*_CF_ = 1.1 Hz), 47.6 (d, ^2^*J*_CF_ = 18.8 Hz) ppm; (minor isomer) δ = 139.2, 130.9 (q, ^2^*J*_CF_ = 32.6 Hz), 128.9, 126.3 (q, ^3^*J*_CF_ = 3.6 Hz), 123.9 (q, ^1^*J*_CF_ = 272.2 Hz), 122.7, 119.7, 111.8 (d, ^1^*J*_CF_ = 244.2 Hz), 109.8, 109.1, 47.7 (d, ^2^*J*_CF_ = 18.6 Hz) ppm; ^19^F NMR (376 MHz, CDCl_3_): (major isomer) δ = −65.89 (s, 1F), −155.34 (dd, *J* = 49.8, 28.3 Hz, 3F) ppm; (minor isomer) δ = −65.86 (s, 1F), -153.71 (dd, *J* = 49.9, 28.6 Hz, 3F) ppm. HRMS (ESI-TOF) *m*/*z*: [M − H]^−^ calcd. for C_13_H_10_F_4_N_2_O_2_ 301.0606; found 301.0612.

**Methyl 4-(2-fluoro-2-nitro-1-(1H-pyrrol-2-yl)ethyl)benzoate** (**3j**). Eluent: Hex/DCM 1:1, DCM; 138 mg (95%), *dr* = 42:58); yellowish oil; ^1^H NMR (400 MHz, CDCl_3_): (major isomer) δ = 8.46 (br s, 1H), 7.93 (d, *J* = 8.3 Hz, 2H), 7.30 (d, *J* = 8.2 Hz, 2H), 6.78–6.74 (m, 1H), 6.31 (dd, ^2^*J*_HF_ = 49.8, 3.4 Hz, 1H), 6.27–6.31 (m, 1H), 6.23 (dd, *J* = 5.9, 2.8 Hz, 1H), 5.03 (dd, ^3^*J*_HF_ = 28.2, 3.4 Hz, 1H), 3.88 (s, 3H) ppm; (minor isomer) δ = 8.46 (br s, 1H), 8.00 (d, *J* = 8.3 Hz, 2H), 7.37 (d, *J* = 8.2 Hz, 2H), 6.78–6.74 (m, 1H), 6.22 (dd, ^2^*J*_HF_ = 49.9, 3.9 Hz, 1H), 6.20–6.16 (m, 2H), 5.05 (dd, ^3^*J*_HF_ = 28.3, 3.8 Hz, 1H), 3.91 (s, 3H) ppm; ^13^C NMR (100 MHz, CDCl_3_): (major isomer) δ = 166.8, 138.51, 130.32, 130.17, 129.27, 124.84, 119.22, 111.14 (d, ^1^*J*_CF_ = 245.8 Hz), 109.04, 108.02 (d, ^4^*J*_CF_ = 1.1 Hz), 52.38, 47.69 (d, ^2^*J*_CF_ = 18.7 Hz) ppm; (minor isomer) δ = 166.7, 140.3 (d, ^3^*J*_CF_ = 0.9 Hz), 130.4, 130.3, 128.5, 122.9, 119.5, 111.8 (d, ^1^*J*_CF_ = 244.1 Hz), 109.5, 108.9, 52.4, 47.8 (d, ^2^*J*_CF_ = 18.8 Hz) ppm. ^19^F NMR (376 MHz, CDCl_3_): (major isomer) δ = −153.04 (dd, *J* = 49.8, 28.2 Hz) ppm; (minor isomer) δ = −151.51 (dd, *J* = 49.9, 28.3 Hz) ppm. HRMS (ESI-TOF) *m*/*z*: [M + H]^+^ calcd. for C_14_H_14_FN_2_O_4_ 293.0932; found 293.0936.

**4-(2-Fluoro-2-nitro-1-(1H-pyrrol-2-yl)ethyl)benzonitrile** (**3k**). Eluent: Hex/DCM 1:1, DCM; 130 mg (85 %); *dr* = 38:62; yellowish oil; ^1^H NMR (400 MHz, CDCl_3_): (major isomer) δ = 8.36 (br s, 1H), 7.61–7.55 (m, 2H), 7.37 (d, *J* = 8.2 Hz, 2H), 6.81–6.76 (m, 1H), 6.31 (dd, ^2^*J*_HF_ = 49.7, 3.3 Hz, 1H), 6.32–6.28 (m, 1H), 6.23 (dd, *J* = 6.0, 2.8 Hz, 1H), 5.06 (dd, ^3^*J*_HF_ = 28.4, 3.2 Hz, 1H) ppm; (minor isomer) δ = 8.36 (br s, 1H), 7.67–7.62 (m, 2H), 7.45 (d, *J* = 8.3 Hz, 2H), 6.81–6.76 (m, 1H), 6.22 (dd, ^2^*J*_HF_ = 49.8, 3.7 Hz, 1H), 6.22–6.17 (m, 2H), 5.09 (dd, ^3^*J*_HF_ = 28.6, 3.6 Hz, 1H) ppm. ^13^C NMR (100 MHz, CDCl_3_): (major isomer) δ = 138.9, 132.7, 130.0, 124.2, 119.5, 118.4, 112.4, 110.8 (d, ^1^*J*_CF_ = 246.2 Hz), 109.2, 108.4 (d, ^4^*J*_CF_ = 1.3 Hz), 47.6 (d, ^2^*J*_CF_ = 18.5 Hz) ppm; (minor isomer) δ = 140.7, 133.0, 129.3, 122.2, 119.7, 118.3, 112.2, 111.3 (d, ^1^*J*_CF_ = 244.5 Hz), 109.8, 109.1, 47.8 (d, ^2^*J*_CF_ = 17.1 Hz) ppm. ^19^F NMR (376 MHz, CDCl_3_): (major isomer) δ = −153.37 (dd, *J* = 49.7, 28.4 Hz) ppm; (minor isomer) δ = −151.89 (dd, *J* = 49.8, 28.6 Hz) ppm. HRMS (ESI-TOF) *m*/*z*: [M + H]^+^ calcd. for C_13_H_11_FN_3_O_2_ 260.0830; found 260.0829.

**2-(2-Fluoro-2-nitro-1-(4-nitrophenyl)ethyl)-1H-pyrrole** (**3l**). Eluent: Hex/DCM 1:1, DCM; 428 mg (>99% yield), scaled-up 1303 mg (99%); *dr* = 40:60; brown waxy solid. ^1^H NMR (400 MHz, CDCl_3_): (major isomer) δ = 8.24 (br s, 1H), 8.14 (d, *J* = 8.6 Hz, 2H), 7.43 (d, *J* = 8.6 Hz, 1H), 6.84–6.79 (m, 1H), 6.33 (dd, ^2^*J*_HF_ = 49.7, 3.2 Hz, 2H), 6.35–6.29 (m, 1H), 6.24 (dd, *J* = 5.7, 2.8 Hz, 1H), 5.12 (dd, ^3^*J*_HF_ = 28.2, 3.1 Hz, 1H) ppm; (minor isomer) δ = 8.31 (br s, 1H), 8.21 (d, *J* = 8.7 Hz, 1H), 6.84–6.79 (m, 1H), 6.25 (dd, ^2^*J*_HF_ = 49.9, 3.7 Hz, 3H), 6.22–6.17 (m, 2H), 5.13 (dd, ^3^*J*_HF_ = 28.4, 3.7 Hz, 1H) ppm. ^13^C NMR (100 MHz, CDCl_3_): (major isomer) δ = 148.0, 140.7, 130.3, 124.1, 124.1, 119.7, 110.9 (d, ^1^*J*_CF_ = 246.1 Hz), 109.4, 108.6 (d, ^4^*J*_CF_ = 1.3 Hz), 47.5 (d, ^2^*J*_CF_ = 18.9 Hz) ppm; (minor isomer) δ = 147.9, 142.5, 129.5, 124.4, 122.1, 119.9, 111.3 (d, ^1^*J*_CF_ = 244.4 Hz), 110.1, 109.3, 47.7 (d, ^2^*J*_CF_ = 19.2 Hz) ppm. ^19^F NMR (376 MHz, CDCl_3_): (major isomer) δ = −153.27 (dd, *J* = 49.7, 28.2 Hz) ppm; (minor isomer) δ = −151.61 (dd, *J* = 49.9, 28.4 Hz) ppm. HRMS (ESI-TOF) *m*/*z*: [M + H]^+^ calcd. for C_12_H_11_FN_3_O_4_ 280.0728; found 280.0728.

**2-(2-Fluoro-2-nitro-1-(3-nitrophenyl)ethyl)-1H-pyrrole** (**3m**). Eluent: Hex/DCM 1:1, DCM; 117 mg (85%); *dr* = 39:61; yellow oil. ^1^H NMR (400 MHz, CDCl_3_): (major isomer) δ = 8.37 (br s, 1H), 8.24–8.09 (m, 2H), 7.61 (d, *J* = 7.7 Hz, 1H), 7.51 (t, *J* = 7.9 Hz, 1H), 6.83–6.78 (m, 1H), 6.34 (s, 1H), 6.34 (dd, ^2^*J*_HF_ = 49.6, 3.3 Hz, 1H), 6.26–6.21 (m, 1H), 5.13 (dd, ^3^*J*_HF_ = 28.4, 3.2 Hz, 1H) ppm; (minor isomer) δ = 8.37 (br s, 1H), 8.24–8.09 (m, 2H), 7.68 (d, *J* = 7.8 Hz, 1H), 7.57 (t, *J* = 8.0 Hz, 1H), 6.83–6.78 (m, 1H), 6.26 (dd, ^2^*J*_HF_ = 49.8, 3.6 Hz, 1H), 6.26–6.16 (m, 2H), 5.15 (dd, ^3^*J*_HF_ = 28.8, 3.6 Hz, 1H) ppm. ^13^C NMR (100 MHz, CDCl_3_): (major isomer) δ = 148.3, 135.4, 134.7, 130.2, 124.2, 124.1, 123.8, 119.6, 110.9 (d, ^1^*J*_CF_ = 245.9 Hz), 109.3, 108.5 (d, ^4^*J*_CF_ = 1.6 Hz), 47.3 (d, ^2^*J*_CF_ = 18.6 Hz) ppm; (minor isomer) δ = 148.5, 137.5, 135.7, 130.4, 123.6, 123.3, 122.2, 119.9, 111.4 (d, ^1^*J*_CF_ = 244.3 Hz), 109.9, 109.2, 47.5 (d, ^2^*J*_CF_ = 18.9 Hz) ppm. ^19^F NMR (376 MHz, CDCl_3_): (major isomer) δ = −153.52 (dd, *J* = 49.6, 28.4 Hz) ppm; (minor isomer) δ = −152.26 (dd, *J* = 49.8, 28.8 Hz) ppm. HRMS (ESI-TOF) *m*/*z*: [M − H]^−^ calcd. for C_12_H_9_FN_3_O_4_ 278.0583; found 278.0587.

**2-(2-Fluoro-2-nitro-1-(2-nitrophenyl)ethyl)-1H-pyrrole** (**3n**). Eluent: Hex/DCM 1:1, DCM; 126 mg (89%); *dr* = 56:44; pale brown oil. ^1^H NMR (400 MHz, CDCl_3_): (major isomer) δ = 8.58 (br s, 1H), 7.85 (d, *J* = 8.1 Hz, 1H), 7.69 (d, *J* = 7.9 Hz, 1H), 7.42–7.64 (m, 2H), 6.79 (s, 1H), 6.34 (br s, 1H), 6.34 (dd, ^2^*J*_HF_ = 49.5, 4.3 Hz, 1H), 6.21 (dd, *J* = 5.4, 2.6 Hz, 1H), 5.83 (dd, ^3^*J*_HF_ = 26.6, 3.8 Hz, 1H) ppm; (minor isomer) δ = 8.38 (br s, 1H), 8.01 (d, *J* = 8.1 Hz, 1H), 7.42–7.64 (m, 3H), 6.74 (s, 1H), 6.34 (br s, 1H), 6.42 (dd, *J* = 50.0, 3.0 Hz, 1H), 6.17 (dd, *J* = 5.4, 2.6 Hz, 1H), 5.83 (dd, *J* = 30.2, 3.8 Hz, 1H) ppm. ^13^C NMR (100 MHz, CDCl_3_): (major isomer) δ = 149.5, 133.5, 131.0 (d, ^4^*J*_CF_ = 4.2 Hz), 129.5, 128.6, 125.1, 124.2 (d, ^3^*J*_CF_ = 1.1 Hz), 119.3, 111.0 (d, ^1^*J*_CF_ = 242.4 Hz), 109.1, 108.0 (d, ^4^*J*_CF_ = 1.7 Hz), 41.1 (d, ^2^*J*_CF_ = 18.2 Hz) ppm; (minor isomer) δ = 148.6, 134.0, 132.0, 130.4, 129.5, 125.3, 122.4, 119.3, 110.7 (d, ^1^*J*_CF_ = 244.1 Hz), 109.8, 109.1, 42.9 (d, ^2^*J*_CF_ = 18.1 Hz) ppm. ^19^F NMR (376 MHz, CDCl_3_): (major isomer) δ = −151.14 (dd, *J* = 49.5, 26.6 Hz) ppm; (minor isomer) δ = −155.79 (dd, *J* = 50.0, 30.2 Hz) ppm. HRMS (ESI-TOF) *m*/*z*: [M − H]^−^ calcd. for C_12_H_9_FN_3_O_4_ 278.0583; found 278.0587.

**1,3-Bis(2-fluoro-2-nitro-1-(1H-pyrrol-2-yl)ethyl)benzene** (**3o**). Eluent: Hex/DCM 1:1, DCM; 191 mg (>99%); mixture of four isomers *dr* = 41:59 (two pair of isomers); greenish oil. ^1^H NMR (400 MHz, CDCl_3_): δ = 8.15 (s, 1H), 8.11 (s, 1H), 8.05 (s, 3H), 8.02 (br s, 1H), 7.96 (br s, 2H), 7.12–7.44 (m, 15H), 7.07 (s, 1H), 6.65–6.76 (m, 8H), 6.23 (dd, ^2^*J*_HF_ = 49.5, 3.6 Hz, 1H), 6.04–6.34 (m, 21H), 6.08 (dd, ^2^*J*_HF_ = 50.1, 4.2 Hz, 1H), 6.06 (dd, ^2^*J*_HF_ = 49.8, 4.2 Hz, 1H), 4.95 (dd, *J* = 28.5, 4.6 Hz, 1H), 4.80–5.03 (m, 5H), 4.91 (dd, ^3^*J*_HF_ = 28.1, 4.6 Hz, 1H), 4.90 (dd, ^3^*J*_HF_ = 28.5, 3.1 Hz, 1H) ppm. ^13^C NMR (100 MHz, CDCl_3_): δ = 136.2, 136.2, 136.1, 136.1, 135.9, 135.9, 135.8, 135.8, 134.4, 134.3, 134.0, 134.0, 130.2, 130.1, 130.0, 130.0, 129.8, 129.8, 129.7, 129.4, 129.3, 128.8, 128.8, 128.7, 128.5, 128.4, 128.3, 125.0, 124.9, 123.3, 123.2, 119.4, 119.4, 119.2, 119.2, 119.1, 119.1, 112.1 (d, ^1^*J*_CF_ = 244.2 Hz), 112.0 (d, ^1^*J*_CF_ = 244.1 Hz), 111.3 (d, ^1^*J*_CF_ = 245.9 Hz), 111.2 (d, ^1^*J*_CF_ = 245.8 Hz), 109.3, 109.2, 109.0, 109.0, 108.9, 108.9, 108.8, 107.9, 107.9, 107.8, 47.8 (d, ^2^*J*_CF_ = 18.0 Hz), 47.7 (d, ^2^*J*_CF_ = 18.0 Hz), 47.7 (d, ^2^*J*_CF_ = 18.7 Hz), 47.6 (d, ^2^*J*_CF_ = 18.5 Hz), 47.5 (d, ^2^*J*_CF_ = 18.5 Hz) ppm (the other signals are overlapped). ^19^F NMR (376 MHz, CDCl_3_): δ = −151.12 (dd, *J* = 49.4, 27.7, 2F Hz), −151.12–−151.41 (m, 2F), −153.23–−153.76 (m, 4F) ppm. HRMS (ESI-TOF) *m*/*z*: [M + H]^+^ calcd. for C_18_H_17_F_2_N_4_O_4_ 391.1212; found 391.1206.

General procedure for the conjugate addition of 1-substituted 1*H*-pyrroles to β-fluoro-β-nitrostyrenes. 

In a typical experiment, a mixture of β-fluoro-β-nitrostyrene (0.5 mmol) and pyrrole (0.4–0.5 g) was loaded into a 4-mL vial (vial G4, Anton Paar). The reaction mixture was placed into a microwave reactor and heated at 150–200 °C with stirring until completion (15–24 h). The reaction progress was monitored by TLC or ^19^F NMR analysis. After completion of the reaction, the resulting mixture was separated by column chromatography on silica gel.

**2-(1-(4-Chlorophenyl)-2-fluoro-2-nitroethyl)-1-methyl-1H-pyrrole** (**3p**). Eluent: Hex/DCM 3:1, Hex/DCM 1:1; 114 mg (76%); *dr* = 36:51; yellow oil. ^1^H NMR (400 MHz, CDCl_3_): (major isomer) δ = 7.33–7.27 (m, 2H), 7.10 (d, *J* = 8.4 Hz, 2H), 6.66–6.61 (m, 1H), 6.37–6.32 (m, 1H), 6.18 (t, *J* = 3.2 Hz, 1H), 6.11 (dd, ^2^*J*_HF_ = 49.8, 4.7 Hz, 1H), 4.93 (dd, ^3^*J*_HF_ = 27.1, 2.9 Hz, 1H), 3.33 (s, 3H) ppm; (minor isomer) δ = 7.38–7.32 (m, 2H), 7.22 (d, *J* = 8.5 Hz, 2H), 6.62–6.58 (m, 1H), 6.40–6.36 (m, 1H), 6.29 (dd, *J* = 49.4, 3.0 Hz, 1H), 6.14 (t, *J* = 3.2 Hz, 1H), 4.96 (dd, *J* = 25.6, 4.7 Hz, 1H), 3.36 (s, 3H) ppm. ^13^C NMR (100 MHz, CDCl_3_): (major isomer) δ = 134.8, 131.3, 130.7, 129.3, 126.3, 123.6, 111.2 (d, ^1^*J*_CF_ = 245.6 Hz), 108.1 (d, ^4^*J*_CF_ = 3.4 Hz), 107.4, 46.0 (d, ^2^*J*_CF_ = 19.1 Hz), 34.0 ppm; (minor isomer) δ = 134.6, 133.3 (d, ^3^*J*_CF_ = 1.9 Hz), 130.1, 129.5, 124.4, 123.5, 112.2 (d, ^1^*J*_CF_ = 246.7 Hz), 108.8 (d, ^4^*J*_CF_ = 3.3 Hz), 107.4, 46.0 (d, ^2^*J*_CF_ = 19.1 Hz), 33.9 ppm. ^19^F NMR (282 MHz, CDCl_3_): (major isomer) δ = −153.51 (dd, *J* = 49.8, 27.1 Hz) ppm; (minor isomer) δ = −152.56 (dd, *J* = 49.4, 25.6 Hz) ppm. HRMS (ESI-TOF) *m*/*z*: [M + H]^+^ calcd. for C_13_H_13_ClFN_2_O_2_ 283.0644; found 283.0961.

**2-(1-(4-Chlorophenyl)-2-fluoro-2-nitroethyl)-1-phenyl-1H-pyrrole** (**3q**). Eluent: Hex/DCM 3:1, Hex/DCM 1:1, DCM; 62 mg (35%); *dr* = 55:45; yellow oil. ^1^H NMR (400 MHz, CDCl_3_): (major isomer) δ = 7.31–7.42 (m, 3H), 7.22–7.17 (m, 2H), 7.02–6.94 (m, 2H), 6.92 (d, *J* = 8.4 Hz, 2H), 6.81 (dd, *J* = 2.8, 1.7 Hz, 1H), 6.49–6.53 (m, 1H), 6.34–6.31 (m, 1H), 6.19 (dd, *J* = 50.0, 3.7 Hz, 1H), 4.76 (dd, *J* = 27.1, 3.5 Hz, 1H) ppm; (minor isomer) δ = 7.42–7.31 (m, 3H), 7.29–7.22 (m, 2H), 7.05 (d, *J* = 8.5 Hz, 2H), 7.02–6.94 (m, 2H), 6.77 (dd, *J* = 2.8, 1.7 Hz, 1H), 6.28–6.31 (m, 1H), 6.58 (dd, *J* = 4.3, 3.7 Hz, 1H), 5.97 (dd, *J* = 49.6, 4.4 Hz, 1H), 4.83 (dd, *J* = 28.7, 3.3 Hz, 1H) ppm. ^13^C NMR (100 MHz, CDCl_3_): (major isomer) δ = 139.0, 134.6, 132.0, 130.6, 129.4, 129.0, 128.5, 127.0, 127.0, 124.0, 111.6 (d, ^1^*J*_CF_ = 244.9 Hz), 109.2 (d, ^4^*J*_CF_ = 3.2 Hz), 108.8, 46.0 (d, ^2^*J*_CF_ = 19.1 Hz) ppm; (minor isomer) δ = 139.0, 134.4, 133.9 (d, ^3^*J*_CF_ = 1.4 Hz), 130.1, 129.4, 129.3, 128.4, 127.1, 125.1, 123.8, 112.0 (d, ^1^*J*_CF_ = 247.0 Hz), 109.7 (d, ^4^*J*_CF_ = 3.6 Hz), 108.9, 45.8 (d, ^2^*J*_CF_ = 18.4 Hz) ppm. ^19^F NMR (282 MHz, CDCl_3_): (major isomer) δ = −152.19 (dd, *J* = 50.0, 27.1 Hz), (minor isomer) δ = −153.32 (dd, *J* = 49.6, 28.7 Hz) ppm. HRMS (ESI-TOF) *m*/*z*: [M + H]^+^ calcd. for C_18_H_15_ClFN_2_O_2_ 345.0801; found 345.0799.

**2-(1-(4-Chlorophenyl)-2-fluoro-2-nitroethyl)-1-(4-ethylphenyl)-1H-pyrrole** (**3r**). Eluent: Hex/DCM 5:1, Hex/DCM 3:1; Hex/DCM 1:1; 51 mg (28%); *dr* = 52:32; yellow oil. ^1^H NMR (400 MHz, CDCl_3_): (major isomer) δ = 7.31–7.14 (m, 4H), 6.94 (d, *J* = 8.4 Hz, 2H), 6.90 (d, *J* = 6.8 Hz, 2H), 6.79 (dd, *J* = 2.8, 1.7 Hz, 1H), 6.53–6.45 (m, 1H), 6.33–6.30 (m, 1H), 6.18 (dd, *J* = 50.0, 3.6 Hz, 1H), 4.76 (dd, *J* = 27.5, 3.7 Hz, 1H), 2.70 (q, *J* = 7.6 Hz, 2H), 1.28 (t, *J* = 7.6 Hz, 3H) ppm; (minor isomer) δ = 7.31–7.14 (m, 4H), 7.07 (d, *J* = 8.4 Hz, 2H), 6.88 (d, *J* = 6.9 Hz, 2H), 6.75 (dd, *J* = 2.8, 1.7 Hz, 1H), 6.60–6.54 (m, 1H), 6.30–6.26 (m, 1H), 5.97 (dd, *J* = 49.7, 4.4 Hz, 1H), 4.83 (dd, *J* = 28.8, 4.1 Hz, 1H), 2.70 (q, *J* = 7.7 Hz, 2H), 1.28 (t, *J* = 7.6 Hz, 3H) ppm.^13^C NMR (100 MHz, CDCl_3_): (major isomer) δ = 144.8, 136.5, 134.5, 132.1, 130.7, 128.9, 128.7, 127.1, 126.8, 124.0, 111.6 (d, ^1^*J*_CF_ = 245.1 Hz), 109.0 (d, ^4^*J*_CF_ = 3.1 Hz), 108.5, 45.9 (d, ^2^*J*_CF_ = 19.0 Hz), 28.6, 15.6 ppm; (minor isomer) δ = 144.6, 136.6, 134.3, 134.0, 130.1, 129.3, 128.7, 126.9, 125.2, 123.8, 112.0 (d, ^1^*J*_CF_ = 246.7 Hz), 109.4 (d, ^4^*J*_CF_ = 3.6 Hz), 108.6, 45.8 (d, ^2^*J*_CF_ = 18.5 Hz), 15.6 ppm. ^19^F NMR (376 MHz, CDCl_3_): (major isomer) δ = −152.34 (ddd, *J* = 50.0, 27.5, 2.3 Hz) ppm; (minor isomer) δ = −153.34 (ddd, *J* = 49.7, 28.8, 2.5 Hz) ppm. HRMS (ESI-TOF) *m*/*z*: [M + H]^+^ calcd. for C_20_H_19_ClFN_2_O_2_ 373.1114; found 373.1116.

**2-(1-(4-Chlorophenyl)-2-fluoro-2-nitroethyl)-1-(p-tolyl)-1H-pyrrole** (**3s**). Eluent: Hex/DCM 5:1, Hex/DCM 4:1, Hex/DCM 1:1; 92 mg (51%); *dr* =48:47; yellow oil. ^1^H NMR (400 MHz, CDCl_3_): (major isomer) δ = 7.25–7.19 (m, 2H), 7.16 (d, *J* = 7.3 Hz, 2H), 6.96 (d, *J* = 8.4 Hz, 2H), 6.88 (d, *J* = 6.4 Hz, 2H), 6.78 (dd, *J* = 2.6, 1.7 Hz, 1H), 6.53–6.47 (m, 1H), 6.33–6.30 (m, 1H), 6.17 (dd, *J* = 50.0, 3.6 Hz, 1H), 4.75 (dd, *J* = 27.5, 3.3 Hz, 1H), 2.40 (s, 3H) ppm; (minor isomer) δ = 7.27 (d, *J* = 8.6 Hz, 2H), 7.17 (d, *J* = 7.0 Hz, 2H), 7.08 (d, *J* = 8.4 Hz, 2H), 6.86 (d, *J* = 6.5 Hz, 2H), 6.74 (dd, *J* = 2.6, 1.7 Hz, 1H), 6.60–6.54 (m, 1H), 6.30–6.26 (m, 1H), 5.97 (dd, *J* = 49.7, 4.3 Hz, 1H), 4.83 (dd, *J* = 28.3, 3.5 Hz, 1H), 2.40 (s, 3H) ppm. ^13^C NMR (100 MHz, CDCl_3_): (major isomer) δ = 138.5, 136.3, 134.5, 132.1, 130.7, 129.9, 128.9, 127.0, 126.8, 124.0, 111.6 (d, ^1^*J*_CF_ = 245.0 Hz), 109.0 (d, ^4^*J*_CF_ = 3.1 Hz), 108.6, 45.8 (d, ^2^*J*_CF_ = 18.9 Hz), 21.2 ppm; (minor isomer) δ = 138.38, 136.41, 134.34, 133.94 (d, ^3^*J*_CF_ = 1.6 Hz), 130.08, 129.89, 129.26, 126.86, 125.12, 123.76, 111.96 (d, ^1^*J*_CF_ = 246.7 Hz), 109.48 (d, *J* = 3.7 Hz), 108.66, 45.74 (d, ^2^*J*_CF_ = 18.4 Hz), 21.21 ppm. ^19^F NMR (376 MHz, CDCl_3_): (major isomer) δ = −152.30 (ddd, *J* = 50.0, 27.5, 2.0 Hz) ppm; (minor isomer) δ = −153.45 (ddd, *J* = 49.7, 28.3, 2.2 Hz) ppm. HRMS (ESI-TOF) *m*/*z*: [M + H]^+^ calcd. for C_19_H_17_ClFN_2_O_2_ 359.0957; found 359.0949.

**1-(3-Chloro-4-methoxyphenyl)-2-(1-(4-chlorophenyl)-2-fluoro-2-nitroethyl)-1H-pyrrole** (**3t**). Eluent: Hex/DCM 5:1, Hex/DCM 4:1, Hex/DCM 1:1; 97 mg (46%); *dr* = 53:43; yellow oil. ^1^H NMR (400 MHz, CDCl_3_): (major isomer) δ = 7.26–7.21 (m, 2H), 6.98–6.92 (m, 3H), 6.87–6.84 (m, 2H), 6.74 (dd, *J* = 2.8, 1.7 Hz, 1H), 6.51–6.46 (m, 1H), 6.31–6.28 (m, 1H), 6.19 (dd, *J* = 49.9, 3.6 Hz, 1H), 4.68 (dd, *J* = 27.2, 3.5 Hz, 1H), 3.93 (s, 3H) ppm; (minor isomer) δ = 7.32–7.27 (m, 2H), 7.09 (d, *J* = 8.5 Hz, 2H), 6.98 (d, *J* = 1.7 Hz, 1H), 6.90–6.87 (m, 2H), 6.70 (dd, *J* = 2.8, 1.7 Hz, 1H), 6.58–6.52 (m, 1H), 6.26 (dd, *J* = 4.8, 3.6 Hz, 1H), 5.97 (dd, *J* = 49.6, 4.3 Hz, 1H), 4.77 (dd, *J* = 28.9, 4.2 Hz, 1H), 3.94 (s, 3H) ppm. ^13^C NMR (100 MHz, CDCl_3_): (major isomer) δ = 155.1, 134.7, 131.9, 131.9, 130.6, 129.1, 127.3, 126.4, 124.1, 122.7, 111.8, 111.4 (d, ^1^*J*_CF_ = 245.0 Hz), 109.1 (d, ^4^*J*_CF_ = 3.1 Hz), 108.8, 56.5, 45.9 (d, ^2^*J*_CF_ = 19.0 Hz) ppm; (minor isomer) δ = 155.1, 134.5, 133.7 (d, ^3^*J*_CF_ = 1.1 Hz), 132.0, 130.1, 129.4, 126.5, 125.4, 123.8, 122.7, 111.9, 111.8 (d, ^1^*J*_CF_ = 247.0 Hz), 109.6 (d, ^4^*J*_CF_ = 3.6 Hz), 108.9, 56.4, 45.8 (d, ^2^*J*_CF_ = 18.4 Hz) ppm. ^19^F NMR (376 MHz, CDCl_3_): (major isomer) δ = −152.40 (ddd, *J* = 49.9, 27.2, 2.4 Hz) ppm; (minor isomer) δ = −153.78 (ddd, *J* = 49.6, 28.9, 2.5 Hz) ppm. HRMS (ESI-TOF) *m*/*z*: [M + H]^+^ calcd. for C_19_H_16_Cl_2_FN_2_O_3_ 409.0517; found 409.0522.

**2-(1-(4-Chlorophenyl)-2-fluoro-2-nitroethyl)-1-(3-methoxyphenyl)-1H-pyrrole** (**3u**). Eluent: Hex/DCM 3:1, Hex/DCM 1:1; 153 mg (74%); *dr* = 53:47; yellow oil. ^1^H NMR (400 MHz, CDCl_3_): (major isomer) δ = 7.33–7.19 (m, 3H), 6.99 (d, *J* = 8.4 Hz, 2H), 6.98–6.91 (m, 1H), 6.83 (dd, *J* = 2.7, 1.7 Hz, 1H), 6.67–6.59 (m, 1H), 6.56–6.50 (m, 1H), 6.49 (t, *J* = 2.1 Hz, 1H), 6.36–6.32 (m, 1H), 6.21 (dd, *J* = 50.0, 3.6 Hz, 1H), 4.84 (dd, *J* = 27.2, 3.6 Hz, 1H), 3.70 (s, 3H) ppm; (minor isomer) δ = 7.33–7.19 (m, 3H), 7.12 (d, *J* = 8.4 Hz, 2H), 6.98–6.91 (m, 1H), 6.80 (dd, *J* = 2.7, 1.7 Hz, 1H), 6.67–6.59 (m, 2H), 6.56–6.50 (m, 1H), 6.32–6.29 (m, 1H), 6.00 (dd, *J* = 49.5, 4.2 Hz, 1H), 4.92 (dd, *J* = 29.2, 4.2 Hz, 1H), 3.72 (s, 3H) ppm. ^13^C NMR (100 MHz, CDCl_3_): (major isomer) δ = 160.1, 139.9, 134.4, 132.1, 130.6, 130.0, 128.9, 126.9, 123.8, 119.0, 114.5, 112.3, 111.5 (d, ^1^*J*_CF_ = 244.9 Hz), 109.2 (d, ^4^*J*_CF_ = 3.1 Hz), 108.7, 55.3, 45.8 (d, ^2^*J*_CF_ = 19.1 Hz) ppm; (minor isomer) δ = 160.1, 140.0, 134.3, 133.9, 130.1, 130.0, 129.2, 124.9, 123.6, 119.1, 114.6, 112.3, 111.8 (d, ^1^*J*_CF_ = 246.9 Hz), 109.7 (d, ^4^*J*_CF_ = 3.7 Hz), 108.8, 55.3, 45.7 (d, ^2^*J*_CF_ = 18.4 Hz). ^19^F NMR (376 MHz, CDCl_3_) (major isomer) δ = −152.33 (dd, *J* = 50.0, 27.2 Hz) ppm; (minor isomer) δ = −153.66 (dd, *J* = 49.5, 29.2 Hz) ppm. HRMS (ESI-TOF) *m*/*z*: [M + H]^+^ calcd. for C_19_H_17_ClFN_2_O_3_ 375.0906; 375.0893.

General procedure for base-induced nitrous acid elimination.

In a typical experiment, the selected adduct **3** (0.4–0.5 mmol, 1 mol equiv.) was treated with a solution of DBU (2.0–6.0 mol equiv.) in acetonitrile (3 mL) with vigorous stirring at room temperature overnight (15–18 h). After completion of the reaction (TLC monitoring), the reaction mixture was concentrated under vacuum. The residue was separated by column chromatography.

**Elimination for 3a**. Purification by column chromatography on silica gel with Hex/DCM 1:1 gave 59 mg (76%) of the target product **4a** as a mixture of diastereomers (*Z*:*E* = 46:54) and 11 mg (13 %) of the side product **5a** as a mixture of diastereomers (*Z*:*E* = 57:43).

(***Z***)-**2-(2-Fluoro-1-phenylvinyl)-1H-pyrrole** (*Z*-**4a**). Colorless oil. ^1^H NMR (400 MHz, CDCl_3_): δ = 8.90 (br s, 1H), 7.50–7.32 (m, 5H), 6.91 (dd, *J* = 4.1, 2.6 Hz, 1H), 6.69 (d, ^2^*J*_HF_ = 84.3 Hz, 1H), 6.31–6.27 (m, 1H), 6.16–6.10 (m, 1H) ppm. ^13^C NMR (100 MHz, CDCl_3_): δ = 143.7 (d, ^1^*J*_CF_ = 265.6 Hz), 134.6 (d, ^3^*J*_CF_ = 10.0 Hz), 129.8, 129.7 (d, ^4^*J*_CF_ = 3.7 Hz), 128.6, 127.0 (d, ^3^*J*_CF_ = 3.1 Hz), 119.4 (d, ^6^*J*_CF_ = 4.8 Hz), 118.9 (d, ^2^*J*_CF_ = 3.6 Hz), 111.1 (d, ^4^*J*_CF_ = 4.6 Hz), 108.9 ppm; ^19^F NMR (376 MHz, CDCl_3_) δ = −132.04 (dd, *J* = 84.3, 6.3 Hz) ppm; HRMS (ESI-TOF) *m*/*z*: [M + H]^+^ calcd. for C_12_H_11_FN 188.0870; found 188.0867.

(***E***)-**2-(2-Fluoro-1-phenylvinyl)-1H-pyrrole** (*E*-**4a**). Colorless oil. ^1^H NMR (400 MHz, CDCl_3_) δ = 7.90 (br s, 1H), 7.50–7.32 (m, 5H), 7.12 (d, ^2^*J*_HF_ = 83.2 Hz, 1H), 6.82–6.77 (m, 1H), 6.26 (dd, *J* = 5.7, 2.9 Hz, 2H) ppm. ^13^C NMR (100 MHz, CDCl_3_): δ = 144.7 (d, ^1^*J*_CF_ = 267.3 Hz), 133.8, 129.8, 128.6, 128.3 (d, ^4^*J*_CF_ = 3.5 Hz), 127.2 (d, ^3^*J*_CF_ = 9.4 Hz), 119.0 (d, ^6^*J*_CF_ = 2.4 Hz), 118.9 (d, ^2^*J*_CF_ = 10.5 Hz), 109.3, 107.6 (d, ^4^*J*_CF_ = 2.6 Hz) ppm. ^19^F NMR (376 MHz, CDCl_3_): δ = −133.82 (d, ^2^*J*_HF_ = 83.2 Hz) ppm. HRMS (ESI-TOF) *m*/*z*: [M + H]^+^ calcd. for C_12_H_11_FN 188.0870; found 188.0867.

(***Z***)-**2-(2-Nitro-1-phenylvinyl)-1H-pyrrole** (*Z*-**5a**). Yellow oil. ^1^H NMR (400 MHz, CDCl_3_): δ = 11.93 (br s, 1H), 7.51–7.36 (m, 4H), 7.34–7.24 (m, 2H), 6.97 (s, 1H), 6.38–6.32 (m, 1H), 6.31–6.26 (m, 1H) ppm. ^13^C NMR (100 MHz, CDCl_3_): δ = 138.4, 138.4, 134.1, 129.6, 129.4, 128.8, 128.3, 126.6, 124.8, 112.0 ppm; HRMS (ESI-TOF) *m*/*z*: [M + H]^+^ calcd. for C_12_H_11_N_2_O_2_ 215.0815; found 215.0819.

(***E***)-**2-(2-Nitro-1-phenylvinyl)-1H-pyrrole** (*E*-**5a**). Yellow oil. ^1^H NMR (400 MHz, CDCl_3_): δ = 11.93 (br s, 1H), 7.58 (s, 1H), 7.51–7.36 (m, 4H), 7.34–7.28 (m, 1H), 7.00 (td, *J* = 2.7, 1.4 Hz, 1H), 6.67–6.61 (m, 1H), 6.38–6.32 (m, 1H) ppm. ^13^C NMR (100 MHz, CDCl_3_): δ = 142.8, 141.7, 129.6 129.1, 128.5, 128.4, 127.8, 125.0, 114.8, 112.3 ppm. HRMS (ESI-TOF) *m*/*z*: [M + H]^+^ calcd. for C_12_H_11_N_2_O_2_ 215.0815; found 215.0819.

**Elimination for 3b**. Purification by column chromatography on silica gel with gradient elution (Hex/DCM 1:1, DCM) gave 23 mg (43%) of the target product **4b** as a mixture of diastereomers (*Z*:*E* = 48:52) and 6 mg (10%) of the side product **5b** as a mixture of diastereomers (*Z*:*E* = 75:25).

(***Z***)-**2-(2-Fluoro-1-(4-methoxyphenyl)vinyl)-1H-pyrrole** (*Z*-**4b**). Colorless oil. ^1^H NMR (400 MHz, CDCl_3_): δ = 8.88 (br s, 1H), 7.35 (d, *J* = 8.5 Hz, 2H), 6.94–6.91 (m, 2H), 6.89 (dd, *J* = 4.1, 2.7 Hz, 1H), 6.64 (d, ^2^*J*_HF_ = 84.7 Hz, 1H), 6.27–6.24 (m, 1H), 6.12–6.07 (m, 1H), 3.84 (s, 3H) ppm. ^13^C NMR (100 MHz, CDCl_3_): δ =159.5, 143.4 (d, ^1^*J*_CF_ = 265.3 Hz), 131.0 (d, ^4^*J*_CF_ = 2.9 Hz), 127.4 (d, ^3^*J*_CF_ = 3.1 Hz), 126.8 (d, ^3^*J*_CF_ = 10.2 Hz), 119.30 (d, ^6^*J*_CF_ = 4.8 Hz), 118.26 (d, ^2^*J*_CF_ = 3.4 Hz), 113.98, 110.99 (d, ^4^*J*_CF_ = 4.8 Hz), 108.93, 55.44 ppm.^19^F NMR (376 MHz, CDCl_3_): δ = −132.54 (dd, *J* = 84.7, 6.3 Hz) ppm. HRMS (ESI-TOF) *m*/*z*: [M + H]^+^ calcd. for C_13_H_13_FNO 218.0976; found 218.0973.

(***E***)-**2-(2-Fluoro-1-(4-methoxyphenyl)vinyl)-1H-pyrrole** (*E*-**4b**). Colorless oil. ^1^H NMR (400 MHz, CDCl_3_): δ = 7.97 (br s, 1H), 7.34–7.30 (m, 2H), 7.05 (d, ^2^*J*_HF_ = 83.5 Hz, 1H), 6.94–6.91 (m, 2H), 6.79 (dd, *J* = 4.2, 2.6 Hz, 1H), 6.23 (dd, *J* = 5.8, 2.9 Hz, 2H), 3.84 (s, 3H) ppm. ^13^C NMR (100 MHz, CDCl_3_): δ = 159.7, 144.3 (d, ^1^*J*_CF_ = 266.1 Hz), 130.9 (d, ^4^*J*_CF_ = 4.0 Hz), 127.5 (d, ^3^*J*_CF_ = 9.9 Hz), 126.0, 118.9 (d, ^6^*J*_CF_ = 2.1 Hz), 118.4 (d, ^2^*J*_CF_ = 9.7 Hz), 114.0, 109.2, 107.6 (d, ^4^*J*_CF_ = 2.7 Hz), 55.4 ppm. ^19^F NMR (376 MHz, CDCl_3_): δ = −134.90 (d, *J* = 83.5 Hz) ppm. HRMS (ESI-TOF) *m*/*z*: [M + H]^+^ calcd. for C_13_H_13_FNO 218.0976; found 218.0973.

(***Z***)-**2-(1-(4-Methoxyphenyl)-2-nitrovinyl)-1H-pyrrole** (*Z*-**5b**). ^1^H NMR (400 MHz, CDCl_3_): δ = 11.87 (br s, 1H), 7.39–7.33 (m, 2H), 7.29–7.22 (m, 1H), 7.03–6.90 (m, 3H), 6.38–6.32 (m, 2H), 3.87 (br s, 3H). ^13^C NMR (100 MHz, CDCl_3_) δ = 161.0, 141.6, 131.3, 130.6, 128.0, 126.3, 124.8, 113.8, 111.8, 55.6 ppm. HRMS (ESI-TOF) *m*/*z*: [M − H]^−^ calcd. for 243.0770; found 243.0752.

**Elimination for 3c**. Purification by column chromatography on silica gel with gradient elution (Hex/DCM 3:1; Hex/DCM 2:1; DCM) gave 73 mg (77%) of the target product **4c** as a mixture of diastereomers (*Z*/*E* = 42:58).

(***Z***)-**2-(1-(4-(*Tert*-butyl)phenyl)-2-fluorovinyl)-1H-pyrrole** (*E*-**4c**). Colorless oil. ^1^H NMR (400 MHz, CDCl_3_): δ = 8.86 (br s, 1H), 7.49–7.41 (m, 2H), 7.41–7.33 (m, 2H), 6.93–6.87 (m, 1H), 6.69 (d, ^2^*J*_HF_ = 84.5 Hz, 1H), 6.32– 6.28 (m, 1H), 6.26–6.16 (m, 1H), 1.39 (s, 9H) ppm. ^13^C NMR (100 MHz, CDCl_3_): δ = 151.3, 143.6 (d, ^1^*J*_CF_ = 265.5 Hz), 131.6 (d, ^4^*J*_CF_ = 10.0 Hz), 129.4 (d, ^4^*J*_CF_ = 2.6 Hz), 127.1 (d, ^3^*J*_CF_ = 3.2 Hz), 125.5, 119.2 (d, ^6^*J*_CF_ = 4.7 Hz), 118.6 (d, ^2^*J*_CF_ = 3.7 Hz), 111.1 (d, ^4^*J*_CF_ = 5.0 Hz), 109.0, 34.7, 31.4 ppm; ^19^F NMR (376 MHz, CDCl_3_): δ = −131.10 (dd, *J* = 84.5, 6.1 Hz) ppm. HRMS (ESI-TOF) *m*/*z*: [M + H]^+^ calcd. for C_16_H_19_FN 244.1496; found 244.1500.

(***E***)-**2-(1-(4-(*Tert*-butyl)phenyl)-2-fluorovinyl)-1H-pyrrole** (*E*-**4c**). Colorless oil. ^1^H NMR (400 MHz, CDCl_3_): δ = 7.92 (br s, 1H), 7.49–7.41 (m, 2H), 7.41–7.33 (m, 2H), 7.10 (d, ^2^*J*_HF_ = 83.4 Hz, 1H), 6.82–6.76 (m, 1H), 6.28–6.22 (m, 2H), 1.40 (s, 9H) ppm. ^13^C NMR (100 MHz, CDCl_3_): δ = 151.4, 144.7 (d, ^1^*J*_CF_ = 267.0 Hz), 130.8, 129.4 (d, ^4^*J*_CF_ = 3.8 Hz), 127.4 (d, ^3^*J*_CF_ = 9.9 Hz), 125.5, 118.9 (d, ^6^*J*_CF_ = 2.1 Hz), 118.7 (d, ^2^*J*_CF_ = 9.8 Hz), 109.2, 107.4 (d, ^4^*J*_CF_ = 2.4 Hz), 34.8, 31.5 ppm. ^19^F NMR (376 MHz, CDCl_3_): δ = −132.94 (d, ^2^*J*_HF_ = 83.4 Hz) ppm. HRMS (ESI-TOF) *m*/*z*: [M + H]^+^ calcd. for C_16_H_19_FN 244.1496; found 244.1500.

**Elimination for 3d**. Purification by column chromatography on silica gel with gradient elution (Hex; Hex/DCM 5:1; Hex/DCM 2:1; Hex/DCM 1:1) gave 29 mg (40%) of the *Z*-isomer and 31 mg (42%) of the *E*-isomer of the target product **4d** and 13 mg (15%) of the side product **5d** as a mixture of diastereomers (*Z*/*E* = 49:51).

**(*Z*)-2-(2-Fluoro-1-(p-tolyl)vinyl)-1H-pyrrole** (*Z*-**4d**). Colorless oil. ^1^H NMR (400 MHz, CDCl_3_): δ = 8.87 (br s, 1H), 7.29 (d, *J* = 8.0 Hz, 2H), 7.19 (d, *J* = 7.9 Hz, 2H), 6.90–6.86 (m, 1H), 6.64 (d, ^2^*J*_HF_ = 84.6 Hz, 1H), 6.24–6.20 (m, 1H), 6.12–6.06 (m, 1H), 2.39 (s, 3H) ppm. ^13^C NMR (100 MHz, CDCl_3_): δ = 2.39 (s, 3H), 143.5 (d, ^1^*J*_CF_ = 265.2 Hz), 138.2, 131.6 (d, ^3^*J*_CF_ = 10.0 Hz), 129.7 (d, ^4^*J*_CF_ = 2.5 Hz), 129.3, 127.2 (d, ^3^*J*_CF_ = 3.3 Hz), 119.3 (d, ^6^*J*_CF_ = 4.8 Hz), 118.6 (d, ^2^*J*_CF_ = 3.3 Hz), 111.0 (d, ^4^*J*_CF_ = 4.7 Hz), 108.9, 21.4 ppm. ^19^F NMR (376 MHz, CDCl_3_): δ = −132.35 (dd, *J* = 84.6, 6.3 Hz) ppm. HRMS (ESI-TOF) *m*/*z*: [M + H]^+^ calcd. for C_13_H_13_FN 202.1027; found 202.1024.

**(*E*)-2-(2-Fluoro-1-(p-tolyl)vinyl)-1H-pyrrole** (*E*-**4d**). Colorless oil. ^1^H NMR (400 MHz, CDCl_3_): δ = 7.92 (br s, 1H), 7.31 (d, *J* = 8.0 Hz, 2H), 7.22 (d, *J* = 7.9 Hz, 2H), 7.08 (d, ^2^*J*_HF_ = 83.4 Hz, 1H), 6.79 (dd, *J* = 3.9, 2.5 Hz, 1H), 6.28–6.24 (m, 1H), 6.23 (dd, *J* = 5.7, 2.8 Hz, 1H), 2.39 (s, 3H) ppm. ^13^C NMR (100 MHz, CDCl_3_): δ = 144.5 (d, ^1^*J*_CF_ = 266.4 Hz), 138.2, 130.8, 129.6 (d, ^4^*J*_CF_ = 3.6 Hz), 129.3, 127.4 (d, ^3^*J*_CF_ = 9.6 Hz), 118.9 (d, ^6^*J*_CF_ = 1.9 Hz), 118.8 (d, ^2^*J*_CF_ = 10.0 Hz), 109.2, 107.4 (d, ^4^*J*_CF_ = 2.6 Hz), 21.4 ppm. ^19^F NMR (376 MHz, CDCl_3_): δ = −134.25 (d, ^2^*J*_HF_ = 83.4 Hz) ppm. HRMS (ESI-TOF) *m*/*z*: [M + H]^+^ calcd. for C_13_H_13_FN 202.1027; found 202.1024.

**2-(2-Nitro-1-(p-tolyl)vinyl)-1H-pyrrole** (**5d**). Pale brown oil. ^1^H NMR (400 MHz, CDCl_3_): δ = 11.91 (br s, 1H), 8.21 (br s, 1H), 7.56 (s, 1H), 7.34–7.09 (m, 8H), 7.01–6.96 (m, 1H), 6.97 (s, 1H), 6.80–6.71 (m, 1H), 6.70–6.62 (m, 1H), 6.39–6.29 (m, 3H), 2.43 (s, 3H), 2.42 (s, 3H) ppm. ^13^C NMR (100 MHz, CDCl_3_): δ = 21.4, 21.6, 111.9, 111.9, 112.2, 114.4, 124.6, 125.0, 126.4, 127.9, 128.3, 128.6, 129.0, 129.5, 129.7, 129.9, 135.5, 139.7, 139.9, 141.9 ppm. HRMS (ESI-TOF) *m*/*z*: [M + H]^+^ calcd. for 229.0972; found 229.0972.

**Elimination for 3e**. Purification by column chromatography on silica gel with Hex/DCM 1:1 gave 63 mg (70%) of the target product **4e** as a mixture of diastereomers (*Z*/*E* = 42:58) and 10 mg (10 %) of the side product **5e** as a mixture of diastereomers (*Z*/*E* = 91:9).

(***Z***)-**2-(2-Fluoro-1-(4-fluorophenyl)vinyl)-1H-pyrrole** (*Z*-**4e**). Colorless oil. ^1^H NMR (400 MHz, CDCl_3_): (*Z*-isomer) δ = 8.93 (br s, 1H), 7.44–7.35 (m, 2H), 7.14–7.06 (m, 2H), 6.94–6.89 (m, 1H), 6.65 (d, *J* = 84.1 Hz, 1H), 6.25–6.22 (m, 1H), 6.06–6.02 (m, 1H) ppm. ^13^C NMR (100 MHz, CDCl_3_): (*Z*-isomer) δ = 162.5 (d, ^1^*J*_CF_ = 248.1 Hz), 143.5 (d, ^1^*J*_CF_ = 265.4 Hz), 131.5 (dd, ^3^*J*_CF_ = 6.0, ^4^*J*_CF_ = 3.2 Hz), 130.5 (dd, ^3^*J*_CF_ = 10.4, ^4^*J*_CF_ = 3.3 Hz), 127.1 (d, ^3^*J*_CF_ = 3.1 Hz), 119.7 (d, ^6^*J*_CF_ = 5.0 Hz), 118.0 (d, ^2^*J*_CF_ = 4.1 Hz), 115.5 (d, ^2^*J*_CF_ = 21.5 Hz), 111.1 (d, ^4^*J*_CF_ = 4.3 Hz), 109.0 ppm. ^19^F NMR (376 MHz, CDCl_3_): δ = −114.24 (tt, *J* = 8.6, 5.4 Hz, 1F), −131.98 (dd, *J* = 84.1, 6.5 Hz, 1F) ppm. HRMS (ESI-TOF) *m*/*z*: [M − H]^−^ calcd. for C_12_H_8_F_2_N 204.0630; found 204.0638.

(***E***)-**2-(2-Fluoro-1-(4-fluorophenyl)vinyl)-1H-pyrrole** (*E*-**4e**). Colorless oil. ^1^H NMR (400 MHz, CDCl_3_): δ = 7.92 (s, 1H), 7.44–7.35 (m, 2H), 7.14–7.06 (m, 2H), 7.08 (d, ^2^*J*_HF_ = 83.1 Hz, 1H), 6.81 (dd, *J* = 4.6, 2.3 Hz, 1H), 6.25–6.22 (m, 2H) ppm. ^13^C NMR (100 MHz, CDCl_3_): δ = 162.53 (d, ^1^*J*_CF_ = 248.1 Hz), 144.67 (d, ^1^*J*_CF_ = 267.6 Hz), 131.4 (dd, ^3^*J*_CF_ = 8.3, ^4^*J*_CF_ = 3.9 Hz), 129.73 (d, ^3^*J*_CF_ = 3.3 Hz), 127.02 (d, ^3^*J*_CF_ = 12.7 Hz), 119.14 (d, ^6^*J*_CF_ = 2.0 Hz), 118.05 (d, ^2^*J*_CF_ = 9.7 Hz), 115.56 (d, ^2^*J*_CF_ = 21.5 Hz), 109.39, 108.06 (d, ^4^*J*_CF_ = 2.8 Hz) ppm. ^19^F NMR (376 MHz, CDCl_3_): δ = −114.75–−114.85 (m), −133.82 (d, ^2^*J*_HF_ = 83.1 Hz) ppm. HRMS (ESI-TOF) *m*/*z*: [M − H]^−^ calcd. for C_12_H_8_F_2_N 204.0630; found 204.0638.

(***Z***)-**2-(1-(4-Fluorophenyl)-2-nitrovinyl)-1H-pyrrole** (*Z*-**5e**). Pale brown oil. ^1^H NMR (400 MHz, CDCl_3_): δ = 11.89 (br s, 1H), 7.45–7.37 (m, 2H), 7.30–7.24 (m, 1H), 7.17–7.08 (m, 2H), 6.93 (s, 1H), 6.38–6.32 (m, 1H), 6.28–6.24 (m, 1H). ^13^C NMR (100 MHz, CDCl_3_): δ = 163.5 (d, ^1^*J*_CF_ = 249.8 Hz), 140.6, 134.4 (d, ^4^*J*_CF_ = 3.5 Hz), 131.6 (d, ^3^*J*_CF_ = 8.3 Hz), 128.4, 127.7, 126.7, 124.9, 115.5 (d, ^2^*J*_CF_ = 21.8 Hz), 112.1 ppm. ^19^F NMR (376 MHz, CDCl_3_) δ = −112.21 (tt, *J* = 8.5, 5.2 Hz) ppm. HRMS (ESI-TOF) *m*/*z*: [M − H]^−^ calcd. for C_12_H_8_FN_2_O_2_ 231.0575; found 231.0572.

**Elimination for 3f**. Purification by column chromatography on silica gel with gradient elution (Hex; Hex/DCM 6:1; Hex/DCM 4:1; Hex/DCM 1:1) gave 43 mg (40%) of the *Z*-isomer and 49 mg (45%) of the E-isomer of the target product **4f** and 18 mg (15%) of the side product **5f** as a mixture of diastereomers (*Z*/*E* = 92:8).

**(*Z*)-2-(1-(4-Bromophenyl)-2-fluorovinyl)-1H-pyrrole** (*Z*-**4f**). Colorless oil. ^1^H NMR (400 MHz, CDCl_3_): δ = 8.92 (br s, 1H), 7.56–7.49 (m, 2H), 7.32–7.26 (m, 2H), 6.94–6.89 (m, 1H), 6.64 (d, ^2^*J*_HF_ = 83.9 Hz, 1H), 6.26–6.20 (m, 1H), 6.06–6.00 (m, 1H) ppm. ^13^C NMR (100 MHz, CDCl_3_): δ = 143.5 (d, ^1^*J*_CF_ = 266.0 Hz), 133.6 (d, ^3^*J*_CF_ = 10.3 Hz), 131.8, 131.4 (d, ^4^*J*_CF_ = 2.6 Hz), 126.6 (d, ^3^*J*_CF_ = 3.1 Hz), 122.5, 119.8 (d, ^6^*J*_CF_ = 5.0 Hz), 118.1 (d, ^2^*J*_CF_ = 4.2 Hz), 111.2 (d, ^4^*J*_CF_ = 4.2 Hz), 109.0 ppm. ^19^F NMR (376 MHz, CDCl_3_): δ = −131.66 (dd, *J* = 83.9, 6.3 Hz) ppm. HRMS (ESI-TOF) *m*/*z*: [M + H]^+^ calcd. for C_12_H_10_^79^BrFN 265.9975; found 265.9971.

(***E***)-**2-(1-(4-Bromophenyl)-2-fluorovinyl)-1H-pyrrole** (*E*-**4f**). Bluish oil. ^1^H NMR (400 MHz, CDCl_3_): δ = 7.91 (br s, 1H), 7.57–7.54 (m, 2H), 7.29 (d, *J* = 8.4 Hz, 2H), 7.07 (d, ^2^*J*_HF_ = 82.8 Hz, 1H), 6.81 (dd, *J* = 4.5, 2.3 Hz, 1H), 6.24 (t, *J* = 2.3 Hz, 2H) ppm. ^13^C NMR (100 MHz, CDCl_3_): δ = 145.0 (d, ^1^*J*_CF_ = 268.9 Hz), 132.7, 131.8, 131.3 (d, ^4^*J*_CF_ = 4.0 Hz), 126.6 (d, ^3^*J*_CF_ = 9.3 Hz), 122.4, 119.3 (d, ^6^*J*_CF_ = 2.0 Hz), 118.1 (d, ^2^*J*_CF_ = 9.4 Hz), 108.3 (d, ^4^*J*_CF_ = 2.8 Hz), 109.4 ppm. ^19^F NMR (376 MHz, CDCl_3_): δ = −132.42 (d, *J* = 82.8 Hz) ppm. HRMS (ESI-TOF) *m*/*z*: [M + H]^+^ calcd. for C_12_H_10_^81^BrFN 267.9955; found 267.9953.

**(*Z*)-2-(1-(4-Bromophenyl)-2-nitrovinyl)-1H-pyrrole** (*Z*-**5f**). Yellow oil. ^1^H NMR (400 MHz, CDCl_3_): (*Z*-isomer) δ = 11.90 (br s, 1H), 7.60–7.53 (m, 2H), 7.32–7.24 (m, 3H), 6.92 (s, 1H), 6.38–6.32 (m, 1H), 6.29–6.24 (m, 1H) ppm. ^13^C NMR (100 MHz, CDCl_3_): (*Z*-isomer) δ = 140.4, 137.3, 131.6, 131.2, 128.2, 127.4, 126.9, 124.9, 124.0, 112.1 ppm. HRMS (ESI-TOF) *m*/*z*: [M + H]^+^ calcd. for C_12_H_10_^79^BrN_2_O_2_ 292.9920; found 292.9914.

**Elimination for 3g**. Purification by column chromatography on silica gel with gradient elution (Hex; Hex/DCM 6:1; Hex/DCM 4:1; Hex/DCM 1:1) gave 36 mg (34%) of the *Z*-isomer and 47 mg (45 %) of the *E*-isomer of the target product **4g** and 13 mg (11%) of the side product **5g** as a mixture of diastereomers (*Z*/*E* = 87:13).

**(*Z*)-2-(1-(4-Chlorophenyl)-2-fluorovinyl)-1H-pyrrole** (*Z*-**4g**). Colorless oil. ^1^H NMR (400 MHz, CDCl_3_): δ = 8.93 (br s, 1H), 7.42–7.30 (m, 4H), 6.94–6.90 (m, 1H), 6.65 (d, ^2^*J*_HF_ = 83.9 Hz, 1H), 6.26–6.21 (m, 1H), 6.06–6.02 (m, 1H) ppm. ^13^C NMR (100 MHz, CDCl_3_): δ = 143.5 (d, ^1^*J*_CF_ = 265.8 Hz), 134.3, 133.1 (d, ^3^*J*_CF_ = 10.3 Hz), 131.1 (d, ^4^*J*_CF_ = 2.7 Hz), 128.8, 126.7 (d, ^3^*J*_CF_ = 3.2 Hz), 119.8 (d, ^6^*J*_CF_ = 5.0 Hz), 118.1 (d, ^2^*J*_CF_ = 4.2 Hz), 111.2 (d, ^4^*J*_CF_ = 4.3 Hz), 109.0 ppm. ^19^F NMR (376 MHz, CDCl_3_) δ = −131.75 (dd, *J* = 83.9, 6.3 Hz) ppm. HRMS (ESI-TOF) *m*/*z*: [M − H]^−^ calcd. for C_12_H_8_^35^ClFN 220.0335; found 220.0343.

**(*E*)-2-(1-(4-chlorophenyl)-2-fluorovinyl)-1H-pyrrole** (*E*-**4g**). Colorless oil. ^1^H NMR (400 MHz, CDCl_3_): δ = 7.92 (br s, 1H), 7.41–7.33 (m, 4H), 7.07 (d, ^2^*J*_HF_ = 82.8 Hz, 1H), 6.81 (dd, *J* = 4.6, 2.3 Hz, 1H), 6.24 (t, *J* = 2.4 Hz, 2H) ppm. ^13^C NMR (100 MHz, CDCl_3_): δ = 145.0 (d, ^1^*J*_CF_ = 268.8 Hz), 134.1, 132.3, 131.0 (d, ^4^*J*_CF_ = 4.0 Hz), 128.8, 126.7 (d, ^3^*J*_CF_ = 9.4 Hz), 119.2 (d, ^6^*J*_CF_ = 1.9 Hz), 118.0 (d, ^2^*J*_CF_ = 9.4 Hz), 109.4, 108.2 (d, ^4^*J*_CF_ = 2.8 Hz) ppm. ^19^F NMR (376 MHz, CDCl_3_): δ = −132.62 (d, *J* = 82.9 Hz) ppm. HRMS (ESI-TOF) *m*/*z*: [M − H]^−^ calcd. for C_12_H_8_^35^ClFN 220.0335; found 220.0343.

**(*Z*)-2-(1-(4-Chlorophenyl)-2-nitrovinyl)-1H-pyrrole** (*Z*-**5g**). Orange solid. M.p. = 122–124 °C (Hex). ^1^H NMR (400 MHz, CDCl_3_): δ = 11.90 (br s, 1H), 7.44–7.38 (m, 2H), 7.38–7.33 (m, 2H), 7.30–7.24 (m, 1H), 6.92 (s, 1H), 6.38–6.32 (m, 1H), 6.30–6.22 (m, 1H) ppm. ^13^C NMR (100 MHz, CDCl_3_): δ = 140.4, 136.8, 135.9, 131.0, 130.0, 128.7, 128.3, 126.9, 124.9, 112.1 ppm. HRMS (ESI-TOF) *m*/*z*: [M − H]^−^ calcd. for C_12_H_10_^35^ClN_2_O_2_ 249.0425; found 249.0421.

**Elimination for 3h**. Purification by column chromatography on silica gel with gradient elution (Hex; Hex/DCM 4:1; Hex/DCM 1:1; DCM) gave 70 mg (60%) of the *Z*-isomer and 4 mg (3%) of the *E*-isomer of the target product **4h** and 14 mg (11%) of the side product **5h** as a mixture of diastereomers (*Z*/*E* = 35:65).

**(*Z*)-2-(1-(2,4-Dichlorophenyl)-2-fluorovinyl)-1H-pyrrole** (*Z*-**4h**). Yellowish oil. ^1^H NMR (400 MHz, CDCl_3_): δ = 8.91 (br s, 1H), 7.50 (s, 1H), 7.33–7.28 (m, 2H), 6.94–6.89 (m, 1H), 6.60 (d, ^2^*J*_HF_ = 83.2 Hz, 1H), 6.23–6.18 (m, 1H), 5.83–5.79 (m, 1H) ppm. ^13^C NMR (100 MHz, CDCl_3_): δ = 144.4 (d, ^1^*J*_CF_ = 267.6 Hz), 136.3 (d, ^4^*J*_CF_ = 3.1 Hz), 135.2, 133.6 (d, ^4^*J*_CF_ = 2.5 Hz), 131.6 (d, ^3^*J*_CF_ = 11.8 Hz), 129.9, 127.2, 126.1 (d, ^3^*J*_CF_ = 3.4 Hz), 119.9 (d, ^6^*J*_CF_ = 5.3 Hz), 115.4 (d, ^2^*J*_CF_ = 6.2 Hz), 110.8 (d, ^4^*J*_CF_ = 4.2 Hz), 109.1 ppm.^19^F NMR (376 MHz, CDCl_3_): δ = −130.76 (dd, *J* = 83.2, 6.2 Hz) ppm. HRMS (ESI-TOF) *m*/*z*: [M + H]^+^ calcd. for C_12_H_9_^35^Cl_2_FN 256.0091; found 256.0101.

**(*E*)-2-(1-(2,4-Dichlorophenyl)-2-fluorovinyl)-1H-pyrrole** (*E*-**4h**). Yellowish oil. ^1^H NMR (400 MHz, CDCl_3_): δ = 7.88 (br s, 1H), 7.51 (d, *J* = 2.1 Hz, 1H), 7.31 (dd, *J* = 8.3, 2.0 Hz, 1H), 7.27 (s, 1H), 7.15 (d, ^2^*J*_HF_ = 82.0 Hz, 1H), 6.79–6.75 (m, 1H), 6.21 (dd, *J* = 6.0, 2.8 Hz, 1H), 6.15–6.10 (m, 1H) ppm. ^13^C NMR (100 MHz, CDCl_3_): δ = 144.8 (d, ^1^*J*_CF_ = 264.1 Hz), 135.1, 135.1 (d, ^4^*J*_CF_ = 1.4 Hz), 132.7 (d, ^4^*J*_CF_ = 1.6 Hz), 131.1, 130.0, 127.5, 126.02 (d, ^3^*J*_CF_ = 7.5 Hz), 119.1 (d, ^6^*J*_CF_ = 2.3 Hz), 116.2 (d, ^2^*J*_CF_ = 13.2 Hz), 109.8, 107.2 (d, ^4^*J*_CF_ = 3.8 Hz) ppm. ^19^F NMR (376 MHz, CDCl_3_): δ = −130.75 (d, *J* = 82.0 Hz) ppm. HRMS (ESI-TOF) *m*/*z*: [M + H]^+^ calcd. for C_12_H_9_^35^Cl^37^ClFN 258.0061; found 258.0069.

**2-(1-(2,4-Dichlorophenyl)-2-nitrovinyl)-1H-pyrrole** (**5h**). Red-brown solid; M.p. = 52–54 °C (DCM) with decomposition. ^1^H NMR (400 MHz, CDCl_3_): (*Z*-isomer) δ = 11.87 (br s, 1H), 7.50 (d, *J* = 2.0 Hz, 1H), 7.41–7.22 (m, 3H), 6.82 (s, 1H), 6.40–6.30 (m, 1H), 6.14–6.07 (m, 1H) ppm; (*E*-isomer) δ = 8.49 (br s, 1H), 7.62 (s, 1H), 7.54 (d, *J* = 2.0 Hz, 1H), 7.39–7.32 (m, 1H), 7.18 (d, *J* = 8.2 Hz, 1H), 7.07–7.02 (m, 1H), 6.52–6.46 (m, 1H), 6.38–6.31 (m, 1H) ppm. ^13^C NMR (100 MHz, CDCl_3_): δ = 138.4, 137.0, 135.9, 135.7, 135.2, 134.5, 133.6, 132.1, 131.8, 130.3, 130.0, 129.9, 128.6, 127.5, 127.3, 127.1, 126.9, 126.5, 125.6, 124.2, 116.6, 112.7, 112.5 ppm. HRMS (ESI-TOF) *m*/*z*: [M + H]^+^ calcd. for C_12_H_9_^35^Cl_2_N_2_O_2_ 283.0036; found: 283.0038.

**Elimination for 3i**. Purification by column chromatography on silica gel with gradient elution (Hex; Hex/DCM 3:1; Hex/DCM 1:1; DCM) gave 42 mg (36%) of the *Z*-isomer and 47 mg (41%) of the *E*-isomer of the target product **4i** and 12 mg (7%) of the side product **5i** as a mixture of diastereomers (*Z*/*E* = 85:15).

**(*Z*)-2-(2-Fluoro-1-(4-(trifluoromethyl)phenyl)vinyl)-1H-pyrrole** (*Z*-**4i**). Colorless oil. ^1^H NMR (400 MHz, CDCl_3_): δ = 8.95 (br s, 1H), 7.66 (d, *J* = 8.2 Hz, 2H), 7.54 (d, *J* = 8.0 Hz, 2H), 6.93 (dd, *J* = 4.1, 2.7 Hz, 1H), 6.68 (d, ^2^*J*_HF_ = 83.5 Hz, 1H), 6.26–6.21 (m, 1H), 6.04–5.99 (m, 1H). ^13^C NMR (100 MHz, CDCl_3_): δ = 143.9 (d, ^1^*J*_CF_ = 266.5 Hz), 138.6 (d, ^3^*J*_CF_ = 10.2 Hz), 130.6 (q, ^2^*J*_CF_ = 32.5 Hz), 130.2 (d, ^4^*J*_CF_ = 2.6 Hz), 126.4 (d, ^3^*J*_CF_ = 2.7 Hz), 125.5 (q, ^3^*J*_CF_ = 3.8 Hz), 124.2 (q, ^1^*J*_CF_ = 272.9 Hz), 120.0 (d, ^6^*J*_CF_ = 5.1 Hz), 118.3 (d, ^2^*J*_CF_ = 4.4 Hz), 111.3 (d, ^4^*J*_CF_ = 4.1 Hz), 109.1 ppm. ^19^F NMR (376 MHz, CDCl_3_) δ = −65.78 (s, 3F), −133.27 (dd, *J* = 83.5, 6.5 Hz, 1F). HRMS (ESI-TOF) *m*/*z*: [M − H]^−^ calcd. for C_13_H_8_F_4_N 254.0598; found 254.0601.

**(*E*)-2-(2-Fluoro-1-(4-(trifluoromethyl)phenyl)vinyl)-1H-pyrrole** (*E*-**4i**). Colorless oil. ^1^H NMR (400 MHz, CDCl_3_): δ = 7.92 (br s, 1H), 7.66 (d, *J* = 8.2 Hz, 2H), 7.54 (d, *J* = 8.4 Hz, 2H), 7.12 (d, ^2^*J*_HF_ = 82.5 Hz, 1H), 6.82 (dd, *J* = 4.6, 2.3 Hz, 1H), 6.25 (t, *J* = 2.4 Hz, 2H). ^13^C NMR (100 MHz, CDCl_3_): δ = 145.6 (d, ^1^*J*_CF_ = 270.4 Hz), 137.6, 130.3 (q, ^2^*J*_CF_ = 32.5 Hz), 130.0 (d, ^4^*J*_CF_ = 4.0 Hz), 126.3 (d, ^3^*J*_CF_ = 9.3 Hz), 125.5 (q, ^3^*J*_CF_ = 3.7 Hz), 124.1 (d, ^1^*J*_CF_ = 270.6 Hz), 119.5 (d, ^6^*J*_CF_ = 1.9 Hz), 118.1 (d, ^2^*J*_CF_ = 9.3 Hz), 109.6, 108.6 (d, *J* = 2.9 Hz) ppm. ^19^F NMR (376 MHz, CDCl_3_) δ = δ − 65.87 (s, 3F), −133.67 (d, *J* = 82.5 Hz, 1F) ppm. HRMS (ESI-TOF) *m*/*z*: [M − H]^−^ calcd. for C_13_H_8_F_4_N 254.0598; found 254.0601.

**(*Z*)-(2-Nitro-1-(4-(trifluoromethyl)phenyl)vinyl)-1H-pyrrole** (**5i**). Orange waxy solid. ^1^H NMR (400 MHz, CDCl_3_): δ = 11.92 (br s, 1H), 7.71 (d, *J* = 8.1 Hz, 2H), 7.55 (d, *J* = 8.0 Hz, 2H), 7.31–7.27 (m, 1H), 6.91 (s, 1H), 6.38–6.33 (m, 1H), 6.23–6.18 (m, 1H) ppm. ^13^C NMR (100 MHz, CDCl_3_): δ = 141.9, 139.9, 131.7 (q, ^2^*J*_CF_ = 33.0 Hz), 130.0, 128.3, 127.1, 125.4 (q, ^3^*J*_CF_ = 3.7 Hz), 124.9, 123.9 (q, ^1^*J*_CF_ = 272.4 Hz), 112.3, 112.3 ppm.^19^F NMR (376 MHz, CDCl_3_) δ = −65.96 ppm. HRMS (ESI-TOF) *m*/*z*: [M-H]^−^ calcd. for C_13_H_8_F_3_N_2_O_2_ 281.0543; found 281.0548.

**Elimination for 3j**. Purification by column chromatography on silica gel with gradient elution (Hex; Hex/DCM 3:1; Hex/DCM 2:1; Hex/DCM 1:2; DCM) gave 43 mg (38%) of the *Z*-isomer and 45 mg (39%) of the *E*-isomer of the target product **4j** and 12 mg (9%) of the side product **5j** as a mixture of diastereomers (*Z*/*E* = 68:32).

**(*Z*)-Methyl 4-(2-fluoro-1-(1H-pyrrol-2-yl)vinyl)benzoate** (Z-**4j**). Colorless oil. ^1^H NMR (400 MHz, CDCl_3_): δ = 8.95 (br s, 1H), 8.09–8.02 (m, 2H), 7.52–7.45 (m, 2H), 6.95–6.89 (m, 1H), 6.69 (d, ^2^*J*_HF_ = 83.6 Hz, 1H), 6.27–6.19 (m, 1H), 6.07–6.00 (m, 1H), 3.94 (s, 3H) ppm. ^13^C NMR (100 MHz, CDCl_3_): δ = 166.9, 143.9 (d, ^1^*J*_CF_ = 266.6 Hz), 139.6 (d, ^3^*J*_CF_ = 10.3 Hz), 130.1, 129.8, 129.8 (d, ^4^*J*_CF_ = 2.6 Hz), 126.3 (d, ^3^*J*_CF_ = 2.9 Hz), 119.8 (d, ^6^*J*_CF_ = 5.0 Hz), 118.6 (d, ^2^*J*_CF_ = 4.3 Hz), 111.2 (d, ^3^*J*_CF_ = 4.2 Hz), 109.0, 52.4 ppm.^19^F NMR (376 MHz, CDCl_3_): δ = −131.15 (dd, *J* = 83.6, 6.4 Hz) ppm. HRMS (ESI-TOF) *m*/*z*: [M + H]^+^ calcd. for C_14_H_13_FNO_2_ 246.0925; found 246.0921.

**(*E*)-Methyl 4-(2-fluoro-1-(1H-pyrrol-2-yl)vinyl)benzoate** (*E*-**4j**). Yellowish oil. ^1^H NMR (400 MHz, CDCl_3_): δ = 8.11 (br s, 1H), 7.97–8.05 (m, 2H), 7.48 (d, *J* = 8.2 Hz, 2H), 7.11 (d, ^2^*J*_HF_ = 82.6 Hz, 1H), 6.82 (dd, *J* = 4.4, 2.4 Hz, 1H), 6.21–6.27 (m, 2H), 3.92 (s, 3H) ppm. ^13^C NMR (100 MHz, CDCl_3_): δ = 166.9, 145.6 (d, ^1^*J*_CF_ = 270.7 Hz), 138.7, 129.8, 129.7, 129.6 (d, ^4^*J*_CF_ = 4.0 Hz), 126.4 (d, ^3^*J*_CF_ = 9.4 Hz), 119.4 (d, ^6^*J*_CF_ = 2.0 Hz), 118.4 (d, ^2^*J*_CF_ = 9.0 Hz), 109.4, 108.5 (d, ^3^*J*_CF_ = 2.7 Hz), 52.3 ppm. ^19^F NMR (376 MHz, CDCl_3_): δ = −130.88 (d, ^2^*J*_HF_ = 82.6 Hz) ppm. HRMS (ESI-TOF) *m*/*z*: [M + H]^+^ calcd. for C_14_H_13_FNO_2_ 246.0925; found 246.0921.

**Methyl 4-(2-nitro-1-(1H-pyrrol-2-yl)vinyl)benzoate** (**5j**). Red-orange solid. M.p. = 153–155 °C with decomposition. ^1^H NMR (400 MHz, CDCl_3_): (*Z*-isomer) δ = 11.93 (br s, 1H), 8.12–8.06 (m, 2H), 7.53–7.45 (m, 2H), 7.30–7.26 (m, 1H), 6.93 (s, 1H), 6.37–6.32 (m, 1H), 6.23–6.19 (m, 1H), 3.96 (s, 3H) ppm; (*E*-isomer) δ = 8.41 (s, 1H), 8.17–8.12 (m, 2H), 7.57 (s, 1H), 7.41–7.36 (m, 2H), 7.05–7.01 (m, 1H), 6.56–6.52 (m, 1H), 6.37–6.32 (m, 1H), 3.96 (s, 3H) ppm. ^13^C NMR (100 MHz, CDCl_3_): δ = 52.5, 52.6, 112.2, 112.5, 116.1, 124.9, 125.3, 127.0, 127.3, 127.5, 128.2, 128.5, 129.1, 129.6, 129.7, 130.0, 130.9, 131.2, 139.1, 140.5, 141.8, 142.8, 166.5, 166.7 ppm. HRMS (ESI-TOF) *m*/*z*: [M + H]^+^ calcd. for C_14_H_13_N_2_O_4_ 273.0870; found 273.0873.

**Elimination for 3k**. Purification by column chromatography on silica gel with gradient elution (Hex/DCM 1:1; DCM) gave 88 mg (81%) of the target product **4k** as a mixture of diastereomers (*Z*/*E* = 47:53) and 11 mg (9%) of the side product **5k** as a mixture of diastereomers (*Z*/*E* = 40:60).

**(*Z*)-4-(2-Fluoro-1-(1H-pyrrol-2-yl)vinyl)benzonitrile** (*Z*-**4k**). Greenish solid. M.p. = 97–99 °C (Hex:DCM 1:1). ^1^H NMR (400 MHz, CDCl_3_): δ = 9.00 (br s, 1H), 7.71–7.63 (m, 2H), 7.57–7.48 (m, 2H), 6.96–6.91 (m, 1H), 6.68 (d, ^2^*J*_HF_ = 83.0 Hz, 1H), 6.26–6.18 (m, 1H), 6.00–5.95 (m, 1H) ppm. ^13^C NMR (100 MHz, CDCl_3_): δ = 143.9 (d, ^1^*J*_CF_ = 267.5 Hz), 139.8 (d, ^3^*J*_CF_ = 10.4 Hz), 132.3, 130.4 (d, ^4^*J*_CF_ = 2.6 Hz), 125.7 (d, ^3^*J*_CF_ = 2.9 Hz), 120.2 (d, ^6^*J*_CF_ = 4.9 Hz), 118.6, 118.2 (d, ^2^*J*_CF_ = 5.0 Hz), 112.1, 111.2 (d, ^4^*J*_CF_ = 3.9 Hz), 109.0 ppm. ^19^F NMR (376 MHz, CDCl_3_): (*Z*-isomer) δ = −129.86 (dd, ^2^*J*_HF_ = 82.2, 6.0 Hz) ppm. HRMS (ESI-TOF) *m*/*z*: [M − H]^−^ calcd. for C_13_H_8_FN_2_ 211.0677; found 211.0674.

**(*E*)-4-(2-Fluoro-1-(1H-pyrrol-2-yl)vinyl)benzonitrile** (*E*-**4k**). Greenish solid. M.p. = 97–99 °C (Hex:DCM 1:1). ^1^H NMR (400 MHz, CDCl_3_): δ = 8.11 (br s, 1H), 7.71–7.63 (m, 2H), 7.56–7.49 (m, 2H), 7.11 (d, ^2^*J*_HF_ = 82.2 Hz, 1H), 6.86–6.82 (m, 1H), 6.28–6.17 (m, 2H) ppm. ^13^C NMR (100 MHz, CDCl_3_): δ = 145.9 (d, ^1^*J*_CF_ = 272.2 Hz), 138.9, 132.2, 130.31 (d, ^4^*J*_CF_ = 4.3 Hz), 125.6 (d, ^3^*J*_CF_ = 9.3 Hz), 119.7 (d, ^6^*J*_CF_ = 1.3 Hz), 118.7, 117.9 (d, ^2^*J*_CF_ = 8.5 Hz), 111.6, 109.4, 109.1 (d, ^4^*J*_CF_ = 3.0 Hz) ppm. ^19^F NMR (376 MHz, CDCl_3_): δ = −130.34 (d, ^2^*J*_HF_ = 83.0 Hz) ppm.

**4-(2-Nitro-1-(1H-pyrrol-2-yl)vinyl)benzonitrile** (**5k**). Yellow oil. ^1^H NMR (400 MHz, CDCl_3_): (*Z*-isomer) δ = 11.90 (br s, 1H), 7.80–7.72 (m, 2H), 7.58–7.51 (m, 2H), 7.33–7.28 (m, 1H), 6.88 (s, 1H), 6.39–6.33 (m, 1H), 6.18–6.14 (m, 1H) ppm; (*E*-isomer) δ = 8.52 (br s, 1H), 7.80–7.72 (m, 2H), 7.54 (s, 1H), 7.47–7.40 (m, 2H), 7.09–7.05 (m, 1H), 6.48–6.41 (m, 1H), 6.39–6.33 (m, 1H) ppm. ^13^C NMR (100 MHz, CDCl_3_): δ = 142.9, 140.9, 139.4, 139.4, 132.4, 132.2, 130.4, 129.3, 129.0, 128.2, 127.4, 127.0, 126.8, 125.8, 124.8, 118.4, 118.2, 117.4, 113.6, 112.7, 112.4, 112.4, ppm. HRMS (ESI-TOF) *m*/*z*: [M + H]^+^ calcd. for C_13_H_10_N_3_O_2_ 240.0768; found 240.0771.

**Elimination for 3l**. Purification by column chromatography on silica gel with gradient elution (Hex; Hex/DCM 5:1; Hex/DCM 2:1; Hex/DCM 1:1) gave 25 mg (29%) of the *Z*-isomer and 18 mg (21%) of the *E*-isomer of the target product **4l**.

**(*Z*)-2-(2-Fluoro-1-(4-nitrophenyl)vinyl)-1H-pyrrole** (*Z*-**4l**). Yellow solid. M.p. = 114–115 °C (Hex/DCM 5:1) with decomposition. ^1^H NMR (400 MHz, CDCl_3_): δ = 8.99 (br s, 1H), 8.25 (d, *J* = 8.5 Hz, 2H), 7.59 (d, *J* = 8.4 Hz, 2H), 6.97–6.92 (m, 1H), 6.71 (d, ^2^*J*_HF_ = 82.8 Hz, 1H), 6.26–6.21 (m, 1H), 6.00–5.95 (m, 1H) ppm. ^13^C NMR (100 MHz, CDCl_3_): δ = 148.0, 144.0 (d, ^1^*J*_CF_ = 267.7 Hz), 141.8 (d, ^3^*J*_CF_ = 10.5 Hz), 130.6 (d, ^4^*J*_CF_ = 2.4 Hz), 125.8 (d, ^3^*J*_CF_ = 3.0 Hz), 123.8, 120.3 (d, ^6^*J*_CF_ = 5.0 Hz), 118.1 (d, ^2^*J*_CF_ = 5.3 Hz), 111.4 (d, ^4^*J*_CF_ = 3.7 Hz), 109.2 ppm. ^19^F NMR (376 MHz, CDCl_3_): δ = −130.01 (dd, *J* = 82.8, 6.5 Hz) ppm. HRMS (ESI-TOF) *m*/*z*: [M − H]^−^ calcd. for C_12_H_8_FN_2_O_2_ 231.0575; found 231.0584.

**(*E*)-2-(2-fluoro-1-(4-nitrophenyl)vinyl)-1H-pyrrole** (*E*-**4l**). Pale brown oil. ^1^H NMR (400 MHz, CDCl_3_): δ = 8.23 (d, *J* = 8.7 Hz, 2H), 8.00 (br s, 1H), 7.59 (d, *J* = 8.6 Hz, 2H), 7.13 (d, ^2^*J*_HF_ = 82.0 Hz, 1H), 6.80–6.89 (m, 1H), 6.20–6.28 (m, 2H) ppm. ^13^C NMR (100 MHz, CDCl_3_): δ = 147.5, 146.3 (d, ^1^*J*_CF_ = 273.8 Hz), 140.8, 130.5 (d, ^4^*J*_CF_ = 4.5 Hz), 125.6 (d, ^3^*J*_CF_ = 8.8 Hz), 123.8, 119.8 (d, ^6^*J*_CF_ = 1.4 Hz), 117.7 (d, ^2^*J*_CF_ = 8.5 Hz), 109.7, 109.4 (d, ^4^*J*_CF_ = 3.1 Hz) ppm. ^19^F NMR (376 MHz, CDCl_3_): δ = −128.64 (d, ^2^*J*_HF_ = 82.0 Hz) ppm. HRMS (ESI-TOF) *m*/*z*: [M − H]^−^ calcd. for C_12_H_8_FN_2_O_2_ 231.0575; found 231.0584.

**Elimination for 3m**. Purification by column chromatography on silica gel with gradient elution (Hex; Hex/DCM 3:1; Hex/DCM 1:1; DCM) gave 31 mg (32%) of the *Z*-isomer and 44 mg (46%) of the *E*-isomer of the target product **4m** and 11.6 mg (11%) of the side product **5m** as a mixture of diastereomers (*Z*/*E* = 62:38).

**(*Z*)-2-(2-Fluoro-1-(3-nitrophenyl)vinyl)-1H-pyrrole** (*Z*-**4m**). Yellow oil. ^1^H NMR (400 MHz, CDCl_3_) δ = 9.02 (br s, 1H), 8.33–8.22 (m, 2H), 7.78–7.73 (m, 1H), 7.58 (t, *J* = 7.9 Hz, 1H), 6.96 (dd, *J* = 4.0, 2.6 Hz, 1H), 6.71 (d, ^2^*J*_HF_ = 82.9 Hz, 1H), 6.29–6.17 (m, 1H), 5.94 (s, 1H) ppm. ^13^C NMR (100 MHz, CDCl_3_): δ = 148.4, 144.0 (d, ^1^*J*_CF_ = 266.9 Hz), 136.6 (d, ^3^*J*_CF_ = 10.8 Hz), 135.9 (d, ^4^*J*_CF_ = 2.6 Hz), 129.6, 126.10 (d, ^3^*J*_CF_ = 3.0 Hz), 124.7 (d, ^4^*J*_CF_ = 2.7 Hz), 123.4, 120.4 (d, ^6^*J*_CF_ = 5.2 Hz), 117.8 (d, ^2^*J*_CF_ = 5.4 Hz), 111.4 (d, ^4^*J*_CF_ = 3.6 Hz), 109.2 ppm. ^19^F NMR (376 MHz, CDCl_3_) δ = −130.64 (dd, *J* = 82.9, 6.6 Hz) ppm. HRMS (ESI-TOF) m/z: [M + H]^+^ calcd. for C_12_H_10_FN_2_O_2_ 233.0721; found 233.0730.

**(*E*)-2-(2-Fluoro-1-(3-nitrophenyl)vinyl)-1H-pyrrole** (*E*-**4m**). Yellow oil. ^1^H NMR (400 MHz, CDCl_3_): δ = 8.29 (br s, 1H), 8.18 (dd, *J* = 8.2, 1.3 Hz, 1H), 8.07 (s, 1H), 7.74 (d, *J* = 7.7 Hz, 1H), 7.56 (t, *J* = 8.0 Hz, 1H), 7.13 (d, ^2^*J*_HF_ = 82.1 Hz, 1H), 6.85 (dd, *J* = 4.0, 2.6 Hz, 1H), 6.25 (dd, *J* = 5.8, 2.9 Hz, 1H), 6.22 (s, 1H). ^13^C NMR (100 MHz, CDCl_3_): δ = 148.4, 145.8 (d, ^1^*J*_CF_ = 271.4 Hz), 135.8, 135.7 (d, ^4^*J*_CF_ = 4.0 Hz), 129.5, 125.7 (d, ^3^*J*_CF_ = 8.9 Hz), 124.5 (d, ^4^*J*_CF_ = 4.4 Hz), 123.1, 119.8 (d, ^6^*J*_CF_ = 1.8 Hz), 117.4 (d, ^2^*J*_CF_ = 8.8 Hz), 109.7, 109.3 (d, ^4^*J*_CF_ = 3.4 Hz) ppm. ^19^F NMR (376 MHz, CDCl_3_) δ = −130.84 (d, ^2^*J*_HF_ = 82.1 Hz) ppm. HRMS (ESI-TOF) *m*/*z*: [M + H]^+^ calcd. for C_12_H_10_FN_2_O_2_ 233.0721; found 233.0730.

**2-(2-Nitro-1-(3-nitrophenyl)vinyl)-1H-pyrrole** (**5m**). Yellow oil. ^1^H NMR (400 MHz, CDCl_3_) (*Z*-isomer) δ = 11.91 (br s, 1H), 8.41–8.32 (m, 1H), 8.33–8.27 (m, 1H), 7.71–7.61 (m, 1H), 7.34–7.30 (m, 1H), 6.92 (s, 1H), 6.40–6.34 (m, 1H), 6.21–6.10 (m, 1H) ppm; (*E*-isomer) δ = 8.52 (br s, 1H), 8.41–8.32 (m, 1H), 8.23–8.16 (m, 1H), 7.80–7.73 (m, 2H), 7.56 (s, 1H), 7.12–7.07 (m, 1H), 6.46–6.40 (m, 1H), 6.40–6.34 (m, 1H) ppm. ^13^C NMR (100 MHz, CDCl_3_): δ = 148.3, 148.1, 140.2, 139.9, 138.7, 136.1, 135.4, 134.6, 129.7, 129.6, 129.2, 128.4, 127.5, 127.2, 127.0, 126.0, 124.9, 124.6, 124.5, 124.2, 123.7, 117.8, 112.9, 112.5 ppm. HRMS (ESI-TOF) *m*/*z*: [M + H]^+^ calcd. for C_12_H_10_N_3_O_4_ 260.0666; found 260.0662.

**Elimination for 3o**. Purification by column chromatography on silica gel with gradient elution (Hex; Hex/DCM 3:1; Hex/DCM 1:1; Hex/DCM 1:2; DCM) gave 21 mg (15%) of the *Z*,*Z*-isomer, 42 mg (29%) of the *E*,*Z*-isomer and 24 mg (17 %) of the *E*,*E*-isomer of the target product **4o** and 13 mg (8 %) of the mixture of (*Z*-NO_2_-*Z*-F)/(*E*-NO_2_-*Z*-F) = 73:27 and 11 mg (7%) of the mixture of (*Z*-NO_2_-*E*-F)/(*E*-NO_2_-*E*-F) = 81:19 of the side product **5o**.

**1,3-Bis((*Z*)-2-fluoro-1-(1H-pyrrol-2-yl)vinyl)benzene** (Z,Z-**4o**). Colorless oil. ^1^H NMR (400 MHz, CDCl_3_) δ 8.93 (br s, 2H), 7.47–7.37 (m, 4H), 6.91 (dd, *J* = 4.0, 2.6 Hz, 2H), 6.68 (d, ^2^*J*_HF_ = 84.1 Hz, 2H), 6.23 (dd, *J* = 5.3, 2.6 Hz, 2H), 6.10–6.05 (m, 2H) ppm. ^13^C NMR (100 MHz, CDCl_3_): δ = 143.8 (d, ^1^*J*_CF_ = 265.7 Hz), 135.0 (d, ^3^*J*_CF_ = 10.2 Hz), 131.0 (d, ^4^*J*_CF_ = 2.7 Hz), 129.7 (d, ^4^*J*_CF_ = 2.6 Hz), 128.7, 126.9 (d, ^3^*J*_CF_ = 3.1 Hz), 119.6 (d, ^6^*J*_CF_ = 5.0 Hz), 118.6 (d, ^2^*J*_CF_ = 3.6 Hz), 111.2 (d, ^4^*J*_CF_ = 4.3 Hz), 109.0. ^19^F NMR (376 MHz, CDCl_3_): δ = −131.92 (dd, *J* = 84.1, 6.5 Hz). HRMS (ESI-TOF) *m*/*z*: [M + H]^+^ calcd. for C_18_H_15_F_2_N_2_ 297.1198; found 297.1197.

**1-((*E*)-2-Fluoro-1-(1H-pyrrol-2-yl)vinyl)-3-((*Z*)-2-fluoro-1-(1H-pyrrol-2-yl)vinyl)benzene** (*E*,*Z*-**4o**). Colorless oil. ^1^H NMR (400 MHz, CDCl_3_) δ = 8.92 (br s, 1H), 7.95 (br s, 1H), 7.48 (s, 1H), 7.46–7.35 (m, 3H), 7.10 (d, ^2^*J*_HF_ = 83.0 Hz, 1H), 6.91 (dd, *J* = 4.0, 2.6 Hz, 1H), 6.80 (dd, *J* = 4.0, 2.5 Hz, 1H), 6.68 (d, ^2^*J*_HF_ = 84.1 Hz, 1H), 6.29–6.20 (m, 3H), 6.15–6.09 (m, 1H). ^13^C NMR (100 MHz, CDCl_3_) δ 144.9 (d, ^1^*J*_CF_ = 267.9 Hz), 143.8 (d, ^1^*J*_CF_ = 265.7 Hz), 134.9 (d, ^3^*J*_CF_ = 10.2 Hz), 134.1, 130.8 (t, ^4^*J*_CF_ = 3.2 Hz), 129.7 (d, ^4^*J*_CF_ = 2.5 Hz), 129.5 (d, ^4^*J*_CF_ = 3.5 Hz), 128.7, 126.9 (d, ^3^*J*_CF_ = 9.6 Hz), 126.9 (d, ^3^*J*_CF_ = 3.3 Hz), 119.6 (d, ^6^*J*_CF_ = 5.0 Hz), 119.2 (d, ^6^*J*_CF_ = 2.0 Hz), 118.7 (d, ^2^*J*_CF_ = 9.2 Hz), 118.6 (d, ^2^*J*_CF_ = 3.5 Hz), 111.1 (d, ^4^*J*_CF_ = 4.5 Hz), 109.4, 109.0, 108.0 (d, ^4^*J*_CF_ = 2.7 Hz). ^19^F NMR (376 MHz, CDCl_3_) δ −131.77 (dd, *J* = 84.1, 6.5 Hz), −133.23 (d, ^2^*J*_HF_ = 83.0 Hz) ppm. HRMS (ESI-TOF) *m*/*z*: [M + H]^+^ calcd. for C_18_H_15_F_2_N_2_ 297.1198; found 297.1197.

**1,3-Bis((*E*)-2-fluoro-1-(1H-pyrrol-2-yl)vinyl)benzene** (*E*,*E*-**4o**). Colorless solid. M.p. = 140-141 ºC with decomposition. ^1^H NMR (400 MHz, CDCl_3_): δ = 7.96 (br s, 2H), 7.52–7.33 (m, 4H), 7.10 (d, ^2^*J*_HF_ = 83.1 Hz, 2H), 6.79 (dd, *J* = 4.1, 2.6 Hz, 2H), 6.27–6.23 (m, 2H), 6.22 (dd, *J* = 5.9, 2.8 Hz, 2H) ppm. ^13^C NMR (100 MHz, CDCl_3_): δ = 144.82 (d, ^1^*J*_CF_ = 267.3 Hz), 133.99, 130.81 (t, ^4^*J*_CF_ = 3.8 Hz), 129.54 (d, ^4^*J*_CF_ = 3.4 Hz), 128.75, 126.92 (d, ^3^*J*_CF_ = 9.4 Hz), 119.14 (d, ^6^*J*_CF_ = 2.1 Hz), 118.74 (d, ^2^*J*_CF_ = 9.8 Hz), 109.38, 107.71 (d, ^4^*J*_CF_ = 2.6 Hz) ppm. ^19^F NMR (376 MHz, CDCl_3_) δ = −133.77 (d, ^2^*J*_HF_ = 83.1 Hz) ppm. HRMS (ESI-TOF) *m*/*z*: [M + H]^+^ calcd. for C_18_H_15_F_2_N_2_ 297.1198; found 297.1197.

**2-((*Z*)-1-(3-((*Z*)-2-fluoro-1-(1H-pyrrol-2-yl)vinyl)phenyl)-2-nitrovinyl)-1H-pyrrole** (*Z*-NO_2_-*Z*-F-**5o**) and **2-((E)-1-(3-((Z)-2-fluoro-1-(1H-pyrrol-2-yl)vinyl)phenyl)-2-nitrovinyl)-1H-pyrrole** (*E*-NO_2_-*Z*-F-**5o**). Yellow oil. ^1^H NMR (400 MHz, CDCl_3_): (major) δ = 11.93 (br s, 1H), 8.98 (br s, 1H), 7.55–7.48 (m, 2H), 7.48–7.42 (m, 2H), 7.29–7.24 (m, 1H), 6.98 (s, 1H), 6.93 (dd, *J* = 4.1, 2.7 Hz, 1H), 6.69 (d, ^2^*J*_HF_ = 83.8 Hz, 1H), 6.39–6.33 (m, 1H), 6.34–6.28 (m, 1H), 6.26–6.19 (m, 1H), 6.06–5.98 (m, 1H) ppm; (minor) δ = 8.87 (br s, 1H), 8.34 (br s, 1H), 7.56 (s, 1H), 7.48–7.42 (m, 2H), 7.39–7.37 (s, 1H), 7.34–7.29 (m, 1H), 7.03 (dd, *J* = 3.9, 2.7 Hz, 1H), 6.90 (dd, *J* = 4.1, 2.7 Hz, 1H), 6.71 (d, ^2^*J*_HF_ = 83.6 Hz, 1H), 6.64–6.59 (m, 1H), 6.39–6.33 (m, 1H), 6.26–6.19 (m, 1H), 6.18–6.14 (m, 1H). ^13^C NMR (100 MHz, CDCl_3_) δ = 145.2, 143.8 (d, ^1^*J*_CF_ = 266.1 Hz), 142.6, 142.4, 141.1, 139.9 (d, ^1^*J*_CF_ = 245.0 Hz), 135.1 (d, ^3^*J*_CF_ = 10.4 Hz), 134.9 (d, ^3^*J*_CF_ = 10.4 Hz), 134.3, 130.9, 130.8 (d, ^4^*J*_CF_ = 2.7 Hz), 130.6 (d, ^4^*J*_CF_ = 2.1 Hz), 130.0 (d, ^4^*J*_CF_ = 2.9 Hz), 129.5, 129.1, 129.0, 128.5, 128.4, 128.2, 128.9, 127.7, 126.7, 126.7, 126.5 (d, ^3^*J*_CF_ = 2.9 Hz), 125.1, 124.9, 119.9 (d, ^6^*J*_CF_ = 5.1 Hz), 119.7 (d, ^2^*J*_CF_ = 4.7 Hz), 118.4 (d, ^2^*J*_CF_ = 3.9 Hz), 115.7, 112.5, 112.1, 111.3 (d, ^4^*J*_CF_ = 5.1 Hz), 111.2 (d, ^4^*J*_CF_ = 4.0 Hz), 109.2, 109.0 ppm. ^19^F NMR (376 MHz, CDCl_3_): (major) δ = −131.57 (dd, *J* = 83.8, 6.7 Hz) ppm; (minor) δ = −131.01 (dd, *J* = 83.6, 5.9 Hz) ppm. HRMS (ESI-TOF) *m*/*z*: [M + H]^+^ calcd. for C_18_H_15_FN_3_O_2_ 324.1143; found 324.1140.

**2-((*Z*)-1-(3-((*E*)-2-fluoro-1-(1H-pyrrol-2-yl)vinyl)phenyl)-2-nitrovinyl)-1H-pyrrole** (*Z*-NO_2_-*E*-F-**5o**) and **2-((*E*)-1-(3-((*E*)-2-fluoro-1-(1H-pyrrol-2-yl)vinyl)phenyl)-2-nitrovinyl)-1H-pyrrole** (*E*-NO_2_-*E*-F-**5o**). Yellow oil. ^1^H NMR (400 MHz, CDCl_3_): (major) δ = 11.92 (br s, 1H), 8.01 (br s, 1H), 7.54–7.47 (m, 2H), 7.45 (t, *J* = 7.6 Hz, 1H), 7.39 (dt, *J* = 7.6, 1.5 Hz, 1H), 7.30–7.25 (m, 1H), 7.10 (d, ^2^*J*_HF_ = 82.7 Hz, 1H), 6.97 (s, 1H), 6.84–6.78 (m, 1H), 6.39–6.31 (m, 2H), 6.23 (t, *J* = 2.3 Hz, 1H), 6.20 (dd, *J* = 6.0, 2.7 Hz, 1H) ppm; (minor) δ = 8.38 (br s, 1H), 8.08 (br s, 1H), 7.59 (d, *J* = 7.8 Hz, 1H), 7.56 (s, 1H), 7.54–7.47 (m, 1H), 7.35–7.30 (m, 1H), 7.30–7.25 (m, 1H), 7.14 (d, ^2^*J*_HF_ = 83.2 Hz, 1H), 7.05–7.00 (m, 1H), 6.84–6.78 (m, 1H), 6.67 (ddd, *J* = 3.9, 2.7, 1.3 Hz, 1H), 6.39–6.31 (m, 1H), 6.31–6.26 (m, 1H), 6.23 (t, *J* = 2.3 Hz, 1H) ppm. ^13^C NMR (100 MHz, CDCl_3_): δ = 146.3, 145.2 (d, ^1^*J*_CF_ = 269.1 Hz), 143.7, 139.9 (d, ^1^*J*_CF_ = 261.7 Hz), 134.2 (d, ^3^*J*_CF_ = 5.1 Hz), 134.0, 131.0, 130.9, 130.7 (d, ^4^*J*_CF_ = 4.1 Hz), 130.7 (d, ^4^*J*_CF_ = 3.5 Hz), 129.6, 129.6, 129.4, 129.1, 129.0, 128.4, 128.4, 128.1, 127.8, 127.7, 126.7, 126.6 (d, ^3^*J*_CF_ = 9.0 Hz), 125.3, 125.0, 119.6 (d, ^6^*J*_CF_ = 2.5 Hz), 119.4 (d, ^6^*J*_CF_ = 2.0 Hz), 118.3 (d, ^2^*J*_CF_ = 9.1 Hz), 115.3, 112.5, 112.1, 109.5, 109.2, 108.6 (d, ^4^*J*_CF_ = 3.0 Hz), 107.5 (d, ^4^*J*_CF_ = 2.0 Hz). ^19^F NMR (376 MHz, CDCl_3_) (major) δ = −132.42 (d, ^2^*J*_HF_ = 82.7 Hz) ppm; (minor) δ = −133.65 (d, ^2^*J*_HF_ = 83.2 Hz) ppm. HRMS (ESI-TOF) *m*/*z*: [M + H]^+^ calcd. for C_18_H_15_FN_3_O_2_ 324.1143; found 324.1140.

**Elimination for 3p**. Purification by column chromatography on silica gel with gradient elution (Hex/DCM 2:1; DCM) gave 58 mg (63%) of the target product **4p** as a mixture of diastereomers (*Z*/*E* = 32:54) and 15 mg (15%) of the side product **5p** as a mixture of diastereomers (*Z*/*E* = 37:63).

**2-(1-(4-Chlorophenyl)-2-fluorovinyl)-1-methyl-1H-pyrrole***(***4p**). Colorless oil. ^1^H NMR (400 MHz, CDCl_3_): (*E*-isomer) δ = 7.34–7.28 (m, 2H), 7.20–7.15 (m, 2H), 7.04 (d, ^2^*J*_HF_ = 82.9 Hz, 1H), 6.75–6.71 (m, 1H), 6.24–6.21 (m, 1H), 6.20 (dd, *J* = 3.6, 1.8 Hz, 1H), 3.36 (d, *J* = 0.8 Hz, 3H) ppm; (*Z*-isomer) δ = 7.38–7.26 (m, 4H), 6.92 (d, ^2^*J*_HF_ = 83.6 Hz, 1H), 6.72–6.69 (m, 1H), 6.25–6.21 (m, 1H), 6.18–6.15 (m, 1H), 3.22 (s, 3H) ppm. ^13^C NMR (100 MHz, CDCl_3_): δ = 147.5 (d, ^1^*J*_CF_ = 278.6 Hz), 146.6 (d, ^1^*J*_CF_ = 267.6 Hz), 135.2 (d, ^3^*J*_CF_ = 6.7 Hz), 133.9, 133.6 (d, ^4^*J*_CF_ = 1.9 Hz), 133.2 (d, ^4^*J*_CF_ = 3.6 Hz), 130.1 (d, ^4^*J*_CF_ = 5.9 Hz), 129.0, 128.8, 128.6 (d, *J* = 3.4 Hz), 127.4 (d, ^3^*J*_CF_ = 12.6 Hz), 125.5, 124.3 (d, ^6^*J*_CF_ = 1.0 Hz), 123.7, 118.1 (d, ^2^*J*_CF_ = 8.6 Hz), 116.6 (d, ^2^*J*_CF_ = 6.2 Hz), 111.5 (d, ^4^*J*_CF_ = 2.5 Hz), 111.3 (d, ^4^*J*_CF_ = 1.3 Hz), 108.0, 107.4, 34.9 (d, ^5^*J*_CF_ = 2.7 Hz), 34.6 ppm. ^19^F NMR (282 MHz, CDCl_3_): (*E*-isomer) δ = −124.05 (d, ^2^*J*_HF_ = 82.9 Hz) ppm; (*Z*-isomer) δ = −122.67 (d, ^2^*J*_HF_ = 83.6 Hz) ppm. HRMS (ESI-TOF) *m*/*z*: [M + H]^+^ calcd. for C_13_H_12_ClFN 236.0637; found 236.0640.

**2-(1-(4-Chlorophenyl)-2-nitrovinyl)-1-methyl-1H-pyrrole** (**5p**). Yellow oil. ^1^H NMR (400 MHz, CDCl_3_) (*E*-isomer) δ = 7.42–7.37 (m, 2H), 7.35 (s, 1H), 7.31–7.27 (m, 2H), 6.91–6.88 (m, 1H), 6.26 (dd, *J* = 3.8, 2.6 Hz, 1H), 6.22 (dd, *J* = 3.8, 1.7 Hz, 1H), 3.38 (s, 3H) ppm; (*Z*-isomer) δ = 7.42–7.37 (m, 2H), 7.32 (s, 1H), 7.25–7.19 (m, 2H), 6.88–6.84 (m, 1H), 6.28 (dd, *J* = 3.9, 1.7 Hz, 1H), 6.19 (dd, *J* = 3.9, 2.7 Hz, 1H), 3.41 (s, 3H) ppm. ^13^C NMR (100 MHz, CDCl_3_): δ = 141.8, 140.7, 137.8, 135.9, 133.9, 133.1, 131.3, 131.1, 130.8, 130.7, 130.7, 129.3, 128.9, 128.5, 127.7, 127.1, 119.1, 116.2, 109.7, 109.4, 36.7, 35.0 ppm. HRMS (ESI-TOF) *m*/*z*: [M + H]^+^ calcd. for C_13_H_12_ClN_2_O_2_ 263.0582; found 263.0583.

**Elimination for 3q**. Purification by column chromatography on silica gel with gradient elution (Hex/DCM 3:1; Hex/DCM 1:1) gave 19 mg (57%) of the target product **4q** as a mixture of diastereomers (*Z*/*E* = 61:31).

**2-(1-(4-Chlorophenyl)-2-fluorovinyl)-1-phenyl-1H-pyrrole** (**4q**). Colorless oil. ^1^H NMR (400 MHz, CDCl_3_) (*Z*-isomer) δ = 7.26–7.06 (m, 7H), 7.04–6.97 (m, 2H), 6.97–6.93 (m, 1H), 6.83 (d, ^2^*J*_HF_ = 82.4 Hz, 1H), 6.41 (dd, *J* = 3.6, 1.8 Hz, 1H), 6.38 (dd, *J* = 3.5, 2.9 Hz, 1H) ppm; (*E*-isomer) δ = 7.26–7.06 (m, 7H), 7.04–6.97 (m, 2H), 6.97–6.93 (m, 1H), 6.94 (d, ^2^*J*_HF_ = 83.4 Hz, 1H), 6.36 (dd, *J* = 3.5, 1.8 Hz, 1H), 6.33–6.30 (m, 1H) ppm. ^13^C NMR (100 MHz, CDCl_3_): δ = 147.3 (d, ^1^*J*_CF_ = 275.6 Hz), 146.65 (d, ^1^*J*_CF_ = 268.1 Hz), 140.3, 140.0, 134.8 (d, ^3^*J*_CF_ = 6.6 Hz), 133.5, 133.1, 133.0 (d, ^3^*J*_CF_ = 2.9 Hz), 130.2 (d, ^4^*J*_CF_ = 5.1 Hz), 128.9, 128.9, 128.6 (d, ^4^*J*_CF_ = 3.3 Hz), 128.5, 128.1, 127.7 (d, ^3^*J*_CF_ = 11.8 Hz), 126.9, 126.8, 125.4, 125.3, 124.9, 124.5 (d, ^6^*J*_CF_ = 1.0 Hz), 123.9, 118.5 (d, ^2^*J*_CF_ = 8.6 Hz), 117.6 (d, ^2^*J*_CF_ = 7.4 Hz), 113.4 (d, ^4^*J*_CF_ = 2.6 Hz), 113.04 (d, ^4^*J*_CF_ = 1.5 Hz), 109.3, 108.7. ^19^F NMR (282 MHz, CDCl_3_) (Z-isomer) δ = −124.11 (d, ^2^*J*_HF_ = 82.4 Hz) ppm; (*E*-isomer) δ = −124.83 (d, ^2^*J*_HF_ = 83.3 Hz) ppm. HRMS (ESI-TOF) *m*/*z*: [M + H]^+^ calcd. for C_18_H_14_ClFN 298.0793; found 298.0786.

**Elimination for 3r**. Purification by column chromatography on silica gel with gradient elution (Hex/DCM 5:1; Hex/DCM 1:1) gave 27 mg (63%) of the target product **4q** as a mixture of diastereomers (*Z*/*E* = 53:32).

**2-(1-(4-Chlorophenyl)-2-fluorovinyl)-1-(4-ethylphenyl)-1H-pyrrole** (**4r**). Yellowish oil. ^1^H NMR (400 MHz, CDCl_3_) (*Z*-isomer) δ = 7.14–6.95 (m, 8H), 6.94–6.90 (m, 1H), 6.82 (d, ^2^*J*_HF_ = 82.4 Hz, 1H), 6.40 (dd, *J* = 3.6, 1.7 Hz, 1H), 6.39–6.36 (m, 1H), 2.59 (q, *J* = 7.7 Hz, 2H), 1.19 (t, *J* = 7.5 Hz, 3H) ppm; (*E*-isomer) δ = 7.14–6.95 (m, 8H), 6.94–6.90 (m, 1H), 6.92 (d, ^2^*J*_HF_ = 83.5 Hz, 1H), 6.34 (dd, *J* = 3.4, 1.8 Hz, 1H), 6.32–6.28 (m, 1H), 2.57 (q, *J* = 7.7 Hz, 2H), 1.17 (t, 3H) ppm. ^13^C NMR (100 MHz, CDCl_3_) δ = 147.2 (d, ^1^*J*_CF_ = 275.0 Hz), 146.6 (d, ^1^*J*_CF_ = 268.1 Hz), 143.2, 143.0, 138.0, 137.7, 134.9 (d, ^3^*J*_CF_ = 6.7 Hz), 133.4, 133.1 (d, ^3^*J*_CF_ = 2.8 Hz), 133.06 (d, ^3^*J*_CF_ = 1.3 Hz), 130.3 (d, ^4^*J*_CF_ = 5.0 Hz), 129.1 (d, ^4^*J*_CF_ = 2.2 Hz), 128.7 (d, ^4^*J*_CF_ = 3.3 Hz), 128.4, 128.2, 128.2, 128.0, 125.5, 125.4, 125.0, 124.5 (d, ^6^*J*_CF_ = 1.1 Hz), 123.8, 118.5 (d, ^2^*J*_CF_ = 8.4 Hz), 117.6 (d, ^2^*J*_CF_ = 7.5 Hz), 113.0 (d, ^4^*J*_CF_ = 2.6 Hz), 112.6 (d, ^4^*J*_CF_ = 1.8 Hz), 109.1, 108.5, 28.5, 28.5, 15.7, 15.7 ppm. ^19^F NMR (376 MHz, CDCl_3_) (*Z*-isomer) δ = −124.17 (d, ^2^*J*_HF_ = 82.5 Hz) ppm; (*E*-isomer) δ = −125.20 (d, ^2^*J*_HF_ = 83.4 Hz) ppm. HRMS (ESI-TOF) *m*/*z*: [M + H]^+^ calcd. for C_20_H_18_ClFN 326.1106; found 326.1097.

**Elimination for 3s**. Purification by column chromatography on silica gel with gradient elution (Hex/DCM 4:1; Hex/DCM 1:1) gave 47 mg (72%) of the target product **4s** as a mixture of diastereomers (*Z*/*E* = 57:35) and 13 mg (18%) of the side product **5s** as a mixture of diastereomers (*Z*/*E* = 41:59).

**2-(1-(4-Chlorophenyl)-2-fluorovinyl)-1-(p-tolyl)-1H-pyrrole** (**4s**). Yellowish oil. ^1^H NMR (400 MHz, CDCl_3_) (*Z*-isomer) δ = 7.18–6.98 (m, 8H), 6.96–6.90 (m, 1H), 6.83 (d, ^2^*J*_HF_ = 82.3 Hz, 1H), 6.41–6.36 (m, 2H), 2.31 (s, 3H) ppm; (*E*-isomer) δ = 7.18–6.98 (m, 8H), 6.96–6.90 (m, 1H), 6.91 (d, ^2^*J*_HF_ = 82.1 Hz, 1H), 6.34 (dd, *J* = 3.3, 1.8 Hz, 1H), 6.31 (t, *J* = 3.1 Hz, 1H), 2.28 (s, 3H) ppm. ^13^C NMR (100 MHz, CDCl_3_): δ = 147.3 (d, ^1^*J*_CF_ = 275.6 Hz), 146.6 (d, ^1^*J*_CF_ = 267.9 Hz), 137.9, 137.5, 136.7, 136.6, 134.9 (d, ^3^*J*_CF_ = 6.7 Hz), 133.5, 133.1, 133.1, 130.2 (d, ^4^*J*_CF_ = 5.1 Hz), 129.5, 129.4, 128.6 (d, ^4^*J*_CF_ = 3.4 Hz), 128.5, 128.1, 127.7 (d, ^3^*J*_CF_ = 12.4 Hz), 125.4, 125.1, 124.7, 124.5 (d, ^6^*J*_CF_ = 1.0 Hz), 123.9, 118.5 (d, ^2^*J*_CF_ = 8.7 Hz), 117.5 (d, ^2^*J*_CF_ = 7.3 Hz), 113.0 (d, ^4^*J*_CF_ = 2.3 Hz), 112.9 (d, ^4^*J*_CF_ = 1.8 Hz), 109.1, 108.5, 21.1, 21.1 ppm. ^19^F NMR (376 MHz, CDCl_3_) (*Z*-isomer) δ = −123.99 (d, ^2^*J*_HF_ = 82.4 Hz) ppm; (*E*-isomer) δ = −124.76 (d, ^2^*J*_HF_ = 83.4 Hz) ppm. HRMS (ESI-TOF) *m*/*z*: [M + O + H]^+^ calcd. for C_19_H_15_ClFNO (adduct with oxygen) 328.0899; found 328.0890.

**2-(1-(4-Chlorophenyl)-2-nitrovinyl)-1-(p-tolyl)-1H-pyrrole** (**5s**). Yellow oil. ^1^H NMR (400 MHz, CDCl_3_) (*E*-isomer) δ = 7.33–7.16 (m, 4H), 7.14–6.96 (m, 5H), 6.75 (s, 1H), 6.32 (dd, *J* = 3.9, 2.7 Hz, 1H), 6.28 (dd, *J* = 3.9, 1.7 Hz, 1H), 2.39 (s, 3H) ppm; (*Z*-isomer) δ = 7.33–7.16 (m, 4H), 7.07 (s, 1H), 7.14–6.96 (m, 5H), 6.42 (d, *J* = 2.2 Hz, 2H), 2.30 (s, 3H) ppm. ^13^C NMR (100 MHz, CDCl_3_) δ = 140.9, 140.7, 138.6, 137.4, 137.3, 137.1, 136.8, 135.6, 135.3, 134.0, 133.5, 132.2, 131.3, 130.6, 130.5, 130.4, 129.9, 129.2, 129.1, 128.4, 127.4, 126.2, 125.9, 124.6, 121.1, 117.3, 110.6, 110.4, 21.2, 21.1 ppm. HRMS (ESI-TOF) *m*/*z*: [M + H]^+^ calcd. for C_19_H_16_ClN_2_O_2_ 339.0895; found 339.0895.

**Elimination for 3t**. Purification by column chromatography on silica gel with gradient elution (Hex/DCM 5:1; Hex/DCM 4:1) gave 46 mg (65%) of the target product **4t** as a mixture of diastereomers (*Z*/*E* = 62:32) and 10 mg (10%) of the side product **5t** as a mixture of diastereomers (*Z*/*E* = 39:61).

**1-(3-Chloro-4-methoxyphenyl)-2-(1-(4-chlorophenyl)-2-fluorovinyl)-1H-pyrrole** (**4t**). Colorless oil. ^1^H NMR (400 MHz, CDCl_3_): (*Z*-isomer) δ = 7.19–7.09 (m, 3H), 7.07–6.91 (m, 3H), 6.89–6.85 (m, 1H), 6.84 (d, ^3^*J*_HF_ = 82.4 Hz, 1H), 6.75 (d, *J* = 6.8 Hz, 1H), 6.40 (dd, *J* = 3.5, 1.7 Hz, 1H), 6.38–6.33 (m, 1H), 3.86 (s, 3H) ppm; (*E*-isomer) δ = 7.19–7.09 (m, 3H), 7.07–6.91 (m, 3H), 6.95 (d, ^3^*J*_HF_ = 83.2 Hz, 1H), 6.89–6.85 (m, 1H), 6.71 (d, *J* = 8.8 Hz, 1H), 6.38–6.33 (m, 1H), 6.29 (t, *J* = 3.2 Hz, 1H), 3.84 (s, 3H). ^13^C NMR (100 MHz, CDCl_3_) δ 153.9, 153.8, 147.3 (d, ^1^*J*_CF_ = 276.0 Hz), 146.8 (d, ^1^*J*_CF_ = 268.6 Hz), 134.7 (d, ^3^*J*_CF_ = 6.6 Hz), 133.7, 133.6, 133.3, 133.3, 132.9 (d, ^3^*J*_CF_ = 3.0 Hz), 130.2 (d, ^4^*J*_CF_ = 5.0 Hz), 128.6, 128.6, 128.2, 127.4, 127.2, 125.6, 124.6, 124.5, 124.3, 123.8, 122.3, 122.3, 118.2 (d, ^2^*J*_CF_ = 8.4 Hz), 117.4 (d, ^2^*J*_CF_ = 7.7 Hz), 113.3 (d, ^4^*J*_CF_ = 2.7 Hz), 112.8 (d, ^4^*J*_CF_ = 1.3 Hz), 111.7, 109.4, 108.8, 56.5, 56.5. ^19^F NMR (376 MHz, CDCl_3_): (*Z*-isomer) δ = −123.66 (d, ^3^*J*_HF_ = 82.4 Hz), (*E*-isomer) δ = −124.46 (d, ^3^*J*_HF_ = 83.2 Hz) ppm. HRMS (ESI-TOF) *m*/*z*: [M + H]^+^ calcd. for C_19_H_15_^35^Cl_2_FNO 362.0509; found 362.0504.

**1-(3-Chloro-4-methoxyphenyl)-2-(1-(4-chlorophenyl)-2-nitrovinyl)-1H-pyrrole** (**5t**). Yellow oil. ^1^H NMR (400 MHz, CDCl_3_) (*E*-isomer) δ = 7.33–7.27 (m, 2H), 7.20–7.16 (m, 2H), 7.13 (d, *J* = 2.6 Hz, 1H), 7.11 (s, 1H), 7.06–6.95 (m, 2H), 6.79 (d, *J* = 8.8 Hz, 1H), 6.45 (dd, *J* = 3.8, 1.7 Hz, 1H), 6.41 (dd, *J* = 3.8, 2.8 Hz, 1H), 3.87 (s, 3H) ppm; (*Z*-isomer) δ = 7.26–7.22 (m, 2H), 7.16 (d, *J* = 2.6 Hz, 1H), 7.06–6.95 (m, 4H), 6.94 (s, 1H), 6.84 (d, *J* = 8.7 Hz, 1H), 6.39 (dd, *J* = 3.9, 1.7 Hz, 1H), 6.32 (dd, *J* = 3.9, 2.8 Hz, 1H), 3.92 (s, 3H) ppm. ^13^C NMR (100 MHz, CDCl_3_) δ = 155.0, 154.3, 141.0, 140.4, 137.5, 135.4, 135.3, 133.9, 133.6, 132.6, 132.3, 131.0, 130.5, 130.4, 129.7, 129.2, 128.4, 128.1, 127.3, 126.9, 126.4, 125.5, 124.3, 123.2, 122.7, 120.0, 117.2, 112.2, 112.0, 110.7, 110.6, 56.6, 56.5 ppm. HRMS (ESI-TOF) *m*/*z*: [M + H]^+^ calcd. for C_19_H_15_^35^Cl_2_N_2_O_3_ 389.0454; found 389.0451.

**Elimination for 3u**. Purification by column chromatography on silica gel with gradient elution (Hex/DCM 2:1; Hex/DCM 1:1; DCM) gave 81 mg (65%) of the target product **4u** as a mixture of diastereomers (*Z*/*E* = 55:35) and 20 mg (15%) of the side product **5u** as a mixture of diastereomers (*Z*/*E* = 44:56).

**2-(1-(4-Chlorophenyl)-2-fluorovinyl)-1-(3-methoxyphenyl)-1H-pyrrole** (**4u**). Yellowish oil. ^1^H NMR (400 MHz, CDCl_3_) (*Z*-isomer) δ = 7.20–7.05 (m, 4H), 7.02 (d, *J* = 8.5 Hz, 1H), 7.00–6.94 (m, 1H), 6.87 (d, ^2^*J*_HF_ = 82.4 Hz, 1H), 6.80–6.65 (m, 3H), 6.44 (dd, *J* = 3.4, 1.6 Hz, 1H), 6.42–6.36 (m, 1H), 3.73 (s, 3H) ppm; (*E*-isomer) δ = 7.20–7.05 (m, 4H), 7.00–6.94 (m, 2H), 6.97 (d, ^2^*J*_HF_ = 83.4 Hz, 1H), 6.80–6.65 (m, 3H), 6.42–6.36 (m, 1H), 6.36–6.31 (m, 1H), 3.71 (s, 3H) ppm. ^13^C NMR (100 MHz, CDCl_3_) δ = 159.8, 147.2 (d, ^1^*J*_CF_ = 275.7 Hz), 146.7 (d, ^1^*J*_CF_ = 268.2 Hz), 141.4, 141.0, 134.7 (d, ^3^*J*_CF_ = 6.6 Hz), 133.4, 133.1, 133.0 (d, ^3^*J*_CF_ = 2.9 Hz), 130.2 (d, ^4^*J*_CF_ = 5.2 Hz), 129.6, 129.6, 128.8 (d, ^3^*J*_CF_ = 3.5 Hz), 128.6 (d, ^4^*J*_CF_ = 3.2 Hz), 128.5, 128.1, 127.7 (d, ^3^*J*_CF_ = 4.1 Hz), 125.3, 124.4, 123.8, 118.5 (d, ^2^*J*_CF_ = 8.4 Hz), 117.6, 117.6 (d, ^2^*J*_CF_ = 7.2 Hz), 117.3, 113.4 (d, ^4^*J*_CF_ = 2.5 Hz), 113.1 (d, ^4^*J*_CF_ = 1.3 Hz), 112.5, 112.4, 111.1, 110.9, 109.3, 108.7, 55.4, 55.4 ppm. ^19^F NMR (376 MHz, CDCl_3_) (*Z*-isomer) δ = −124.12 (d, *J* = 82.4 Hz) ppm; (*E*-isomer) δ = −124.80 (d, ^2^*J*_HF_ = 83.3 Hz) ppm. HRMS (ESI-TOF) *m*/*z*: [M + H]^+^ calcd. for C_19_H_16_^35^ClFNO 328.0899; found 328.0893 ppm.

**2-(1-(4-chlorophenyl)-2-nitrovinyl)-1-(3-methoxyphenyl)-1H-pyrrole** (**5u**). Yellow oil. ^1^H NMR (400 MHz, CDCl_3_): δ = 7.32–7.22 (m, 3H), 7.21–7.06 (m, 4H), 7.11 (s, 1H), 6.79–6.65 (m, 2H), 6.47 (dd, *J* = 3.8, 1.7 Hz, 1H), 6.45–6.40 (m, 1H), 3.73 (s, 3H) ppm; δ = 7.32–7.22 (m, 1H), 7.21–7.06 (m, 3H), 7.04 (dd, *J* = 2.5, 2.0 Hz, 1H), 6.91–6.85 (m, 1H), 6.86 (s, 1H), 6.79–6.65 (m, 3H), 6.37–6.31 (m, 2H), 3.80 (s, 3H) ppm. ^13^C NMR (100 MHz, CDCl_3_) δ = 160.5, 160.1, 140.9, 140.6, 140.6, 140.4, 137.3, 135.4, 135.3, 133.8, 133.7, 132.2, 130.9, 130.6, 130.5, 130.4, 130.0, 129.3, 129.0, 128.3, 127.1, 126.2, 120.6, 118.4, 117.4, 117.1, 113.7, 112.8, 112.2, 110.9, 110.6, 110.4, 55.6, 55.5 ppm. HRMS (ESI-TOF) *m*/*z*: [M + H]^+^ calcd. for C_19_H_16_ClN_2_O_3_ 355.0844; found 355.0831.

**5,5′-((4-Chlorophenyl)methylene)bis(2-(2-fluoro-2-nitro-1-(4-nitrophenyl)ethyl)-1H-pyrrole)** (**6**). A solution of adduct **3l** (90 mg, 2.3 mol. equiv.) and 4-chlorobenzaldehyde (23 mg, 1 mol. equiv.) in DCM (1 mL) was loaded into a vial. Next, TFA (1 drop) was added to the solution. The reaction mixture was stirred overnight at room temperature. After completion of the reaction (TLC monitoring), the reaction mixture was concentrated under vacuum. The product was isolated as a mixture of isomers by column chromatography on silica using DCM as the eluent. Pale brown oil. ^1^H NMR (400 MHz, CDCl_3_) δ = 8.21–8.06 (m, 4H), 8.03–7.85 (m, 2H), 7.52–7.34 (m, 4H), 7.27–7.20 (m, 2H), 7.07–7.00 (m, 2H), 6.41–6.03 (m, 4H), 5.87–5.69 (m, 2H), 5.37–5.30 (m, 1H), 5.13–4.89 (m, 2H) ppm. ^13^C NMR (100 MHz, CDCl_3_) δ = 148.0, 147.8, 142.3, 142.2, 142.2, 140.7, 140.7, 139.5, 139.5, 139.4, 139.4, 133.8, 133.8, 133.7, 133.6, 133.6, 133.3, 133.3, 133.3, 130.2, 130.2, 130.1, 129.7, 129.7, 129.7, 129.4, 129.4, 129.0, 129.0, 129.9, 124.4, 124.2, 124.1, 124.1, 122.2, 122.1, 122.0, 112.3, 111.1 (d, ^1^*J*_CF_ = 246.1 Hz), 111.1 (d, ^1^*J*_CF_ = 244.8 Hz), 110.9 (d, ^1^*J*_CF_ = 246.7 Hz), 110.8, 110.8 (d, ^1^*J*_CF_ = 246.4 Hz), 110.8, 110.7, 110.7, 110.6, 110.6, 109.1, 109.1, 109.0, 108.8, 108.8, 108.7, 47.7 (d, ^2^*J*_CF_ = 17.5 Hz), 47.6 (d, ^2^*J*_CF_ = 17.0 Hz), 47.6 (d, ^2^*J*_CF_ = 18.3 Hz), 47.5 (d, ^2^*J*_CF_ = 19.2 Hz), 43.3, 43.3 (the other signals are overlapped) ppm. ^19^F NMR (376 MHz, CDCl_3_) δ = −150.39–−150.95 (m), −151.00–−151.72 (m) ppm.

## 4. Conclusions

In summary, new synthetic approaches to novel classes of monofluorinated 2-substituted pyrroles were developed. The conjugate addition of pyrroles to β-Fluoro-β-nitrostyrenes was investigated to yield 2-(2-Fluoro-2-nitro-1-arylethyl)-1*H*-pyrroles. The reaction proceeded under catalyst-free neat conditions. It was demonstrated that structurally diverse β-Fluoro-β-nitrostyrenes and pyrroles efficiently participated in this transformation. As a result, a family of novel adducts was synthesized in a quantitative yield. The kinetics of this reaction were studied to evaluate the activation parameters and substituent effects. Furthermore, it was found that 2-(2-Fluoro-2-nitro-1-arylethyl)-1*H*-pyrroles are prone to efficiently undergo the subsequent base-induced elimination of nitrous acid to furnish 2-(2-Fluoro-1-arylvinyl)-1*H*-pyrroles. This approach demonstrated a broad scope in terms of differently substituted nitrostyrenes and pyrroles. The chemo- and stereoselectivity of this process were studied. A family of novel monofluorinated 2-vinylpyrroles was prepared in up to an 85% isolated yield. The compounds herein obtained can be potential building blocks to diverse fluorine-substituted pyrrole derivatives. The preparation of the corresponding dipyrromethane was demonstrated. Thus, the proposed synthetic sequence involving a conjugate addition and HNO_2_ elimination represents a remarkable example of an effective alternative route in which β-Fluoro-β-nitrostyrenes serve as a formal analog of 1-Fluoroacetylenes.

## Data Availability

The data presented in this study are available in Appendix A.

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
