# Peer review of "An Easy Synthesis of Monofluorinated Derivatives of Pyrroles from β-Fluoro-β-Nitrostyrenes"

_molecules, 2021, doi:10.3390/molecules26123515_

Round 1
Reviewer 1 Report
In this research article entitled “Synthesis of Monofluorinated Derivatives of Pyrroles via their Reaction with β-Fluoro-β-Nitrostyrenes”, Valentine G. Nenajdenko and co-workers reported a very interesting synthetic work, where a large family of β-fluoro-β-nitrostyrenes were used as starting reagents in a very easy conjugated Michael addition of pyrroles, performed under catalyst-free and solvent-free conditions, affording the corresponding 2-(2-fluoro-2-nitro-1-arylethyl)-1H-pyrroles. The subsequent base-induced elimination of nitrous acid then afforded a series of 2-(2-fluoro-1-arylvinyl)-1H-pyrroles, which are monofluorinated derivatives of pyrroles with appealing agrochemical and pharmaceutical applications. In my opinion, the research design of the work is appropriate and the results are clearly presented: the paper is certainly interesting for scientists expert in the field of organic synthesis and perfectly fits with the scope of the journal. Therefore, I have no doubts in recommending its publication in the MDPI Molecules journal, after the following minor revisions.
1) First of all, in my opinion the title of the paper should be more incisive and specific: "via their reaction with" is too generic. A possible suggestion is “An Easy Two-Step Synthesis of Monofluorinated Derivatives of Pyrroles from β-Fluoro-β-Nitrostyrenes” or similar ones.
2) Second, in the introduction authors reported several references about the application of pyrrole derivatives in pharmaceuticals, agrochemicals and dyes. However, I think that some reviews describing the synthesis of pyrrole heterocycles could also be added (for example, I suggest the following: DOI: 10.1039/C3CS60015G; DOI: 10.3390/catal10010025).
3) Concerning the three schemes of substrate scope (i.e., Scheme 2, Scheme 4 and Scheme 5), in the captions authors should specify the reaction conditions and how yields and diasteroisomeric ratio (dr) were determined.
4) Other minor issues: a) I suggest to change “in neat” with “under solvent-free conditions”; b) in a part of the paper, compound numbers are not shown in bold; c) On line 76, I think that 3n should be actually 3o; d) in the Table 3, the substrate numbering is in my opinion misleading (as they could be confused with a range of different products): I suggest to change 1b-me with 1b', 1b-h with 1b'' and 1b-cl with 1b'''; e) on line 177, I think that adduct 4l is actually 3l; f) the titles on lines 283 and 296 should be written in bold; g) on line 1141, “1-fluoroacetylenes” should be changed with “1-fluoroacetylenes.”.
Author Response
1) First of all, in my opinion the title of the paper should be more incisive and specific: "via their reaction with" is too generic. A possible suggestion is “An Easy Two-Step Synthesis of Monofluorinated Derivatives of Pyrroles from β-Fluoro-β-Nitrostyrenes” or similar ones.
The title was corrected
2) Second, in the introduction authors reported several references about the application of pyrrole derivatives in pharmaceuticals, agrochemicals and dyes. However, I think that some reviews describing the synthesis of pyrrole heterocycles could also be added (for example, I suggest the following: DOI: 10.1039/C3CS60015G; DOI: 10.3390/catal10010025).
Several references were added including the suggested ones.
3) Concerning the three schemes of substrate scope (i.e., Scheme 2, Scheme 4 and Scheme 5), in the captions authors should specify the reaction conditions and how yields and diasteroisomeric ratio (dr) were determined.
Corrected
4) Other minor issues: a) I suggest to change “in neat” with “under solvent-free conditions”;
Changed
- b) in a part of the paper, compound numbers are not shown in bold;
Corrected
- c) On line 76, I think that 3nshould be actually 3o;
Corrected
- d) in the Table 3, the substrate numbering is in my opinion misleading (as they could be confused with a range of different products): I suggest to change 1b-mewith 1b', 1b-hwith 1b'' and 1b-cl with 1b''';
changed
- e) on line 177, I think that adduct 4lis actually 3l;
corrected
- f) the titles on lines 283 and 296 should be written in bold;
corrected
- g) on line 1141, “1-fluoroacetylenes” should be changed with “1-fluoroacetylenes.”
corrected
Reviewer 2 Report
The paper describes the synthesis of monofluorinated derivatives of pyrroles via their reaction with β-fluoro-β-nitrostyrenes. The products described are new and their characterisation is acceptable. The work is interesting and the presentation is adequate.
I suggest that the paper is accepted after the following corrections are done.
Corrections:
- line 51: "yielding novel 2-(2-fluoro-2-nitro-1-ar-ylethyl)-1H-pyrroles". Please refer to a compound number after numbering these compounds in scheme 1.
- The reaction presented in scheme 2 seems rather wastefull as pyrrole was used in excess (as the solvent). Did the authors attempt the reaction with 1 equiv. of this reagent and if so with what results?
- Scheme 2. "Overnight" is not really a reaction time. Was it 12h or 18h? Please specify the exact reaction time. Also this shows poor reaction execution as it shows that no monitoring of the reaction took place during the night therefore the reactions could have ended anywhere from 2h to 16h or more.
- line 72-74: "The formation of two diastereomers was probably caused by the high acidity of the proton on the carbon bearing the fluorine and nitro groups (the estimated pKa of CH2FNO2 is around 9.5)[27]" Do the authors have any actual proof that a change in the dr is actually taking place? ie. Monitoring the reaction with NMR should show a gradual change in the dr as should leaving the reaction mixture to run for longer under the same conditions. A base is not present in the reaction thereby I am not convinced that this is the problem.
- The authors describe the isolation in all cases of products as mixtures of diastereoisomers. However, no effort was made to separate the isomers and determine the structure of the major diastereoisomer ie. by X-ray crystallography. This is a major drawback of this study.
- Scheme 5: No reaction times are shown. Please edit.
- Line 228: "Due to the reduced nucleophilicity of these substrates". Not all of these substrates have reduced nucleophilicity. The N-methyl pyrrole should have increased nucleophilicity and thereby not require more forcing conditions. Please explain.
- Experimental: It would be preferable to also have the Rf values and the IR data of all new compounds.
- In the supporting information file some 1H NMR spectra can not be viewed clearly. Please provide zoomed images for all regions where the peaks are not clearly visible
Author Response
Corrections:
- line 51: "yielding novel 2-(2-fluoro-2-nitro-1-ar-ylethyl)-1H-pyrroles". Please refer to a compound number after numbering these compounds in scheme 1.
corrected
- The reaction presented in scheme 2 seems rather wastefull as pyrrole was used in excess (as the solvent). Did the authors attempt the reaction with 1 equiv. of this reagent and if so with what results?
Before we tried reactions in water media, but even with 1.5-2.0 equiv. of pyrrole we observed the formation of some amount of product of double addition of nitrostyrene along with the target mono-adduct in reduced total yield (about 65 %). On the contrary, the reaction with large excess of pyrrole demonstrated to be selective and high effective. We used 0.5 ml of pyrrole as comfortable volume to operate with model milligram scale reactions. Moreover, if needed pyrrole can be easily distilled off and recycled. Nevertheless, we admit that it may be possible to minimize the pyrrole consumption to some extent without lowering the yield.
- Scheme 2. "Overnight" is not really a reaction time. Was it 12h or 18h? Please specify the exact reaction time. Also this shows poor reaction execution as it shows that no monitoring of the reaction took place during the night therefore the reactions could have ended anywhere from 2h to 16h or more.
Corrected. The actual time is in range 25-30h. In this case we obtained the kinetic curves that are themselves detailed monitoring of the reaction and allow with sufficient accuracy to estimate the time of full conversion.
- line 72-74: "The formation of two diastereomers was probably caused by the high acidity of the proton on the carbon bearing the fluorine and nitro groups (the estimated pKa of CH2FNO2 is around 9.5) [27]" Do the authors have any actual proof that a change in the dr is actually taking place? ie. Monitoring the reaction with NMR should show a gradual change in the dr as should leaving the reaction mixture to run for longer under the same conditions. A base is not present in the reaction thereby I am not convinced that this is the problem.
According to NMR monitoring the dr was not changing during the reaction since it proceeded under the kinetic control.
- The authors describe the isolation in all cases of products as mixtures of diastereoisomers. However, no effort was made to separate the isomers and determine the structure of the major diastereoisomer ie. by X-ray crystallography. This is a major drawback of this study.
The separation of the isomers by chromatography and the direct identification of them was not possible because they have the same retention time on silica.
- Scheme 5: No reaction times are shown. Please edit.
Corrected. Reaction times are shown in the caption
- Line 228: "Due to the reduced nucleophilicity of these substrates". Not all of these substrates have reduced nucleophilicity. The N-methyl pyrrole should have increased nucleophilicity and thereby not require more forcing conditions. Please explain.
Low reactivity of N-methyl pyrrole as well as N-aryl pyrroles compared to unsubstituted pyrrole can be explained by high steric demand of substituent. Indeed, reaction with N-methyl pyrrole demonstrates negligible conversion even at 110 °C under microwave activation.
- Experimental: It would be preferable to also have the Rf values and the IR data of all new compounds.
Not registered. Because of a lot of routine work and large number of samples we were focused only on basic characterization of new compounds: 1H, 13C, 19F, HRMS to fully confirm the structure..
- In the supporting information file some 1H NMR spectra can not be viewed clearly. Please provide zoomed images for all regions where the peaks are not clearly visible
Corrected
Reviewer 3 Report
I would suggest to improve the Abstract, first sentences really suffer of the lack of concrete information, they are rather general. The authors have to take into account that main audience now is reading only abstracts and to change it from this viewpoint. All other parts of the manuscript - introduction and the main body are very good by chemical content and text preparation
Author Response
I would suggest to improve the Abstract, first sentences really suffer of the lack of concrete information, they are rather general. The authors have to take into account that main audience now is reading only abstracts and to change it from this viewpoint. All other parts of the manuscript - introduction and the main body are very good by chemical content and text preparation
The first sentence was removed. A sentence about kinetics was added
Round 2
Reviewer 2 Report
The paper can now be accepted.